# Clinical and genomic features of Chinese lung cancer patients with germline mutations

Wenying Peng[1,2], Bin Li[3,4], Jin Li[5], Lianpeng Chang[5], Jing Bai[5], Yuting Yi[5], Rongrong Chen[5], Yanyan Zhang[5], Chen Chen[5], Xingxiang Pu[1], Meilin Jiang[1], Jia Li[1], Rui Zhong[1], Fang Xu[1], Bolin Chen[1], Li Xu[1], Ning Wang[6], Jiaojiao Huan[5], Pingping Dai[5], Yanfang Guan[5], Ling Yang[5], Xuefeng Xia[5], Xin Yi[5], Jiayin Wang[7], Fenglei Yu [8,9✉] & Lin Wu [1,9✉]

The germline mutation landscape in Chinese lung cancer patients has not been well defined. In this study, sequencing data of 1,021 cancer genes of 1,794 Chinese lung cancer patients was analyzed. A total of 111 pathogenic or likely pathogenic germline mutations were identified, significantly higher than non-cancer individuals (111/1794 vs. 84/10,588, $p < 2.2e-16$). *BRCA1/2* germline mutations are associated with earlier onset age (median 52.5 vs 60 years-old, $p = 0.008$). Among 29 cancer disposition genes with germline mutations detected in Chinese cohort and/or TCGA lung cancer cohort, Only 11 from 29 genes are identified in both cohorts and *BRCA2* mutations are significantly more common in Chinese cohort ($p = 0.015$). Chinese patients with germline mutations have different prevalence of somatic *KRAS, MET* exon 14 skipping and *TP53* mutations compared to those without. Our findings suggest potential ethnic and etiologic differences between Western and Asian lung cancer patients.

[1] The Second Department of Thoracic Oncology, Hunan Cancer Hospital/the Affiliated Cancer Hospital of Xiangya School of Medicine, Central South University, 410000 Changsha, China. [2] The second department of Oncology, Yunnan Cancer Hospital & The Third Affiliated Hospital of Kunming Medical University & Yunnan Cancer Center, 650000 Kunming, China. [3] Department of Oncology, Xiangya Hospital, Central South University, 410000 Changsha, China. [4] National Clinical Research Center for Geriatric Disorders, Xiangya Hospital, Central South University, 410000 Changsha, China. [5] Geneplus-Beijing, 103306 Beijing, China. [6] Department of Oncology, The PLA Rocket Force Characteristic Medical Center, 100088 Beijing, China. [7] Department of Computer Science and Technology, School of Electronic and Information Engineering, Xi'an Jiaotong University, 710049 Xi'an, China. [8] Department of Thoracic Surgery, The Second Xiangya Hospital of Central South University, 410000 Changsha, China. [9] These authors jointly supervised this work: Fenglei Yu, Lin Wu. ✉email: yufenglei@csu.edu.cn; wulin-calf@yeah.net

Many human cancers could be inheritable. Over 100 genes, mostly tumor suppressor genes, have been identified to be accountable for inheritable cancers, a phenomenon termed, genetic predisposition as exampled by germline mutations in *BRCA1* and *BRCA2* for predisposition of breast cancers, ovary cancers and mismatch repair (MMR) genes for cancers associated with Lynch syndrome[1,2]. In addition, patients with these mutations may have distinct biological and clinical features that are managed differently. For examples, ovary cancer patients with *BRCA1* or *BRCA2* mutations benefit particularly from Poly ADP-ribose polymerase inhibitor Olaparib while solid tumors with MMR mutations have demonstrated high response rate to immune checkpoint blockade[3,4]. However, these well-known genes only account for a small fraction of the genetic burden in cancers and the genetic alterations that may be responsible for predisposition to many potentially inheritable cancers are largely unknown.

Lung cancer is the leading cause of cancer-related death worldwide. It has been long known that a family history of lung cancer is associated with increased risks for lung cancer in both smokers and never smokers[5–7], suggesting the potential genetic predisposition for lung cancer development. Well-defined, high penetrance, hereditary lung-cancer syndromes are uncommon. Recent pan-cancer studies have demonstrated that 3.5–8.5% of lung cancers harbor likely pathogenic germline mutations[8,9]. Several well-known predisposition genetic variants including *BRCA2* and *CHEK2* have been found to have strong association with lung cancer risk[10] and rare pathogenic germline mutations in genes of Fanconi anemia pathway also contribute to the risk of squamous lung cancers[11]. However, all these pioneer studies are based on western populations and the germline mutation landscape in Asian lung cancer patients remains largely unknown. Given the distinct genomic landscape of Asian lung cancer patients[12], it is reasonable to speculate that genetic predisposition variants may be different between Asian lung cancer patients and western counterparts. Furthermore, the current standard of care for treatment of metastatic lung cancers is based upon the determination of actionable somatic driver gene mutations[13]. Recent studies have demonstrated germline mutations can co-occur or be mutually exclusive with somatic cancer gene alterations[8], but little attention has been paid to the somatic mutational landscape in the setting of co-occurring pathogenic germline mutations.

In this study, we analyzed the next generation sequencing (NGS) data of 1021 cancer genes from 1794 Chinese lung cancer patients with the intent to delineate the germline mutational landscape in Chinese lung cancer patients as well as clinical and genomic features of these patients with germline mutations. Pathogenic or likely pathogenic (P/LP) germline mutations of 35 cancer genes were identified in 106 of the 1794 Chinese patients (5.91%). *BRCA1/2* germline mutations are associated with younger age. Prevalence of somatic mutations in *KRAS*, *MET* exon 14 skipping and *TP53* is different in patients with P/LP germline mutations compared to those without.

## Results

**Germline mutation landscape of Chinese lung cancer patients.** Germline DNA and paired tumor DNA were subjected to NGS of 1021 cancer genes with an average sequencing depth of 285× (36×−441×) in germline DNA and 1248× (56×−4626×) in tumor DNA respectively (Fig. 1). Comparison of the single-nucleotide polymorphism (SNP) data from the current cohort to that of individuals submitted to 1000 genomes project phase 3 ($n = 2054$)[14,15] revealed that the mean pairwise F-statistics (Fixation indices, Fst) difference was significant between the

lung cancer patients in the current cohort and African (Fst = 0.07), European (Fst = 0.06), South Asian (Fst = 0.04) and Admixed American (Fst = 0.04) populations; however, the SNP architecture of the lung cancer patients in the current cohort was almost identical to the East Asian (ASN) individuals (Fst = 0.00) (Fig. 2a). Furthermore, the principal component analysis (PCA) using SNP from 1000 genome project also demonstrated that the lung cancer patients from this study were significantly clustered with East Asians but clearly separated from other ethnic populations (Fig. 2b). Taken together, these data suggested that these 1794 Chinese lung cancer patients are likely genetically ordinal Chinese.

A total of 111 pathogenic/likely pathogenic (P/LP) germline mutations from 35 known cancer susceptibility genes were identified in 106 (5.91%) patients according to American College of Medical Genetics and Genomics (ACMG) 2015 guideline[16]. One hundred and one of the 106 patients carried one P/LP germline mutation and five patients harbored two P/LP germline gene mutations (Supplementary data 1). The demographic, clinical and pathological features and the prevalence of germline mutations are shown in Table 1. The most commonly mutated gene in this Chinese lung cancer cohort was *BRCA2* in 14 patients. In addition, *BRCA1* germline mutations were identified in four patients (Fig. 3a). Seventeen of the 18 *BRCA1/2* mutations have been reported in public database (Clinvar or BRCA Share[17]) or previous studies on breast cancers[18]. A novel frameshift mutation, *BRCA2*: c.5163_5164delCA (p.N1721Kfs*5), was identified and defined as a P/LP mutation based on ACMG guideline[16]. Other frequently mutated genes included *FANCA* in nine patients, *RAD51D* in seven patients, *ATM* in seven patients, *MUTYH* in six patients, and *TP53* in five patients, etc. (Supplementary data 1).

To illustrate the potential association between these P/LP germline mutations and lung carcinogenesis in this cohort, we annotated the P/LP mutations from a recently published whole genome sequencing data of non-cancer individuals enrolled in the China Metabolic Analytics Project (ChinaMAP) ($n = 10,588$), a study on the impact of genetic architecture on metabolic diseases[19]. Based on the same 2015 ACMG guideline under the same filtering criteria (see Methods), 84 P/LP germline mutations were identified in the same 35 cancer predisposition genes,

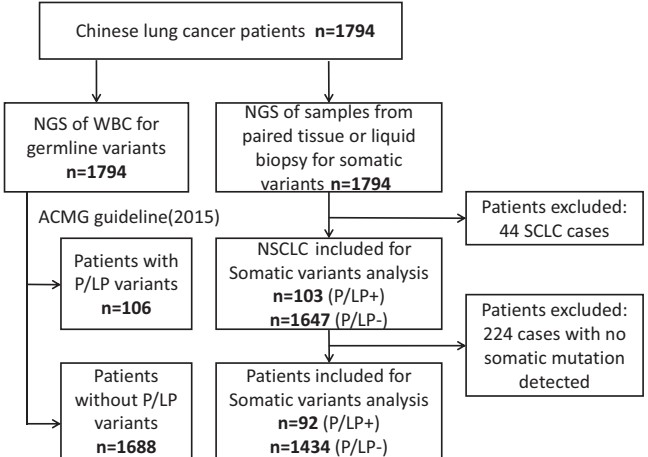

**Fig. 1 Study scheme of patient cohort.** Patients diagnosed with lung caner went through targeted sequencing as part of clinical care from November 2017 to August 2018 were included for further analysis. NGS next generation sequencing. WBC white blood cells. ACMG American college of medical genetics and genomics, NSCLC non-small cell lung cancer, SCLC small cell lung cancer, P/LP pathogenic or likely pathogenic.

a

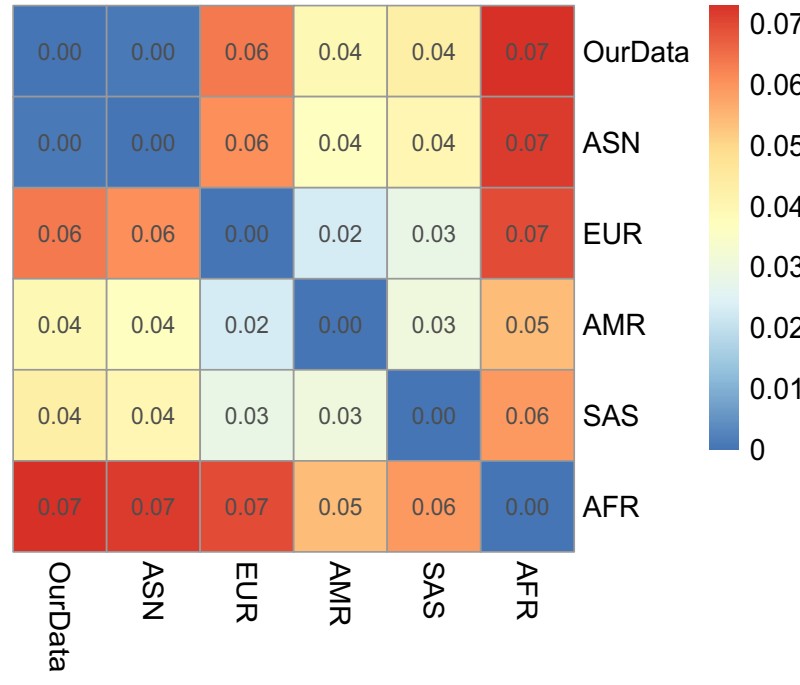

b

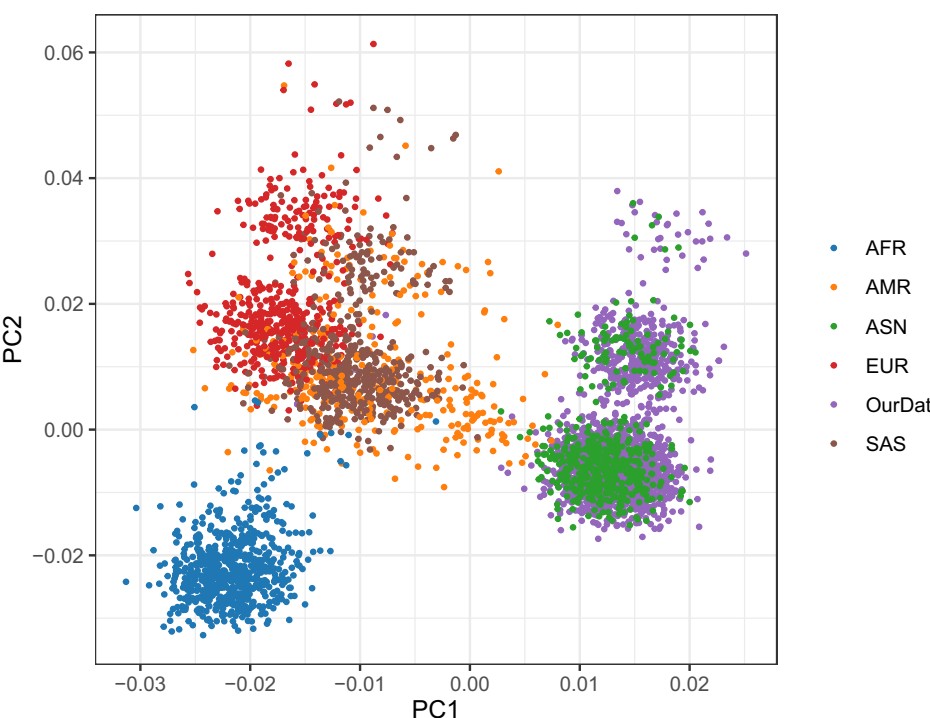

**Fig. 2 Genetic architecture of 1794 lung cancer patients. a** Pairwise Fst difference between the Chinese lung cancer cohort and other populations. **b** Principal component analysis (PCA) of Chinese Lung cancer patients and other populations in 1000 genome project. AFR African. AMR Admixed American. ASN East Asian. EUR European. SAS South Asian.

**Table 1 Clinical characteristics and prevalence of germline mutations.**

|  | Patients With P/LP germline variants | Patients without P/LP germline variants | Prevalence of P/LP germline variants |
|---|---|---|---|
| Number of patients(N) | 106 | 1688 | 5.91% |
| Age at diagnosis—yrs |  |  | $p = 0.28$[a] |
| Mean ± SD | 58.36 ± 11.51 | 59.97 ± 11.54 |  |
| Median (range) | 60.5(29–82) | 60(16–94) |  |
| NA(N) | 2 | 77 |  |
| Gender—N (%) |  |  | $p = 0.56$[b] |
| Female | 42(39.6) | 725(43.0) | 5.48% |
| Male | 64(60.4) | 961(57.0) | 6.24% |
| NA | 0 | 2 |  |
| Histologic diagnosis—N (%) |  |  | $p = 0.50$[b] |
| LUAD | 75(70.7) | 1148(68.0) | 6.13% |
| LUSC | 10(9.4) | 160(9.4) | 5.88% |
| SCLC | 3(2.8) | 41(2.4) | 6.82% |
| Other type | 5(4.7) | 39(2.3) | 11.36% |
| NSCLC-NOS | 13(12.2) | 300(17.7) |  |
| Stage at diagnosis[c]-N (%) |  |  | $p = 0.75$[b] |
| I | 1 (0.9) | 37 (2.2) | 2.63% |
| II | 3 (2.8) | 47 (2.8) | 6.00% |
| III | 8 (7.5) | 153 (9.1) | 4.97% |
| IV | 76 (71.7) | 1034 (61.2) | 6.85% |
| NA | 18 (17.0) | 417 (24.7) |  |

*P/LP* pathogenic/likely pathogenic, *NA* not available, *LUAD* lung adenocarcinoma, *LUSC* lung squamous carcinoma, *SCLC* small cell lung cancer, *NOS* not otherwise specified; Other histology type included: large cell neuroendocrine carcinoma, adenosquamous carcinoma, sarcomatoid carcinoma, pleomorphic carcinoma, poorly differentiated carcinoma, mucoepidermoid carcinoma, lymphoepithelioid carcinoma, etc.
[a]*P* value is calculated with Mann–Whitney test.
[b]*P* value is calculated with Chi-square test or Fisher's exact test.
[c]The clinical stage was based on 8th AJCC non-small cell lung cancer stage edition.

significantly lower than that in the lung cancer cohort (84/10,588 (0.80%) vs. 111/1794 (6.1%), $p < 2.2e-16$, Chi-square test). We then compared the allele frequency (AF) of the germline mutations of each gene in lung cancer patients to that in the non-cancer individuals. There were 17 genes with P/LP germline mutations detected in ≥3 patients in this lung cancer cohort. As shown in the Table 2, 16 of the 17 genes had AF of P/LP germline mutations higher (significantly higher in 11 genes) in the lung cancer patients than in non-cancer individuals in ChinaMAP study indicating an enrichment of these germline mutations in lung cancer patients. Furthermore, of the 106 patients with germline mutations, we were able to collect and analyze tumor samples from 59 patients and loss of heterozygosity (LOH) of the second allele was found in 8 (12.9%) tumors (Supplementary Table 1) and lost-of-function mutations in the other allele were found in three additional tumors (Supplementary Table 2) for a total of 11 (18.6%) patients showing evidence of second-hit events, comparable to that in the western patient population[20]. Taken together, these data suggest that these P/LP germline mutations identified in the current study may be associated with increased risk of lung cancer development in Chinese population.

**Germline landscape between Asian and western lung cancers.** To understand whether germline landscape differs between Asian and western lung cancer population, we compared our results with the germline mutation data derived from the Cancer Genome Atlas (TCGA) lung adenocarcinoma (LUAD), and lung squamous carcinoma (LUSC) cohorts[9]. Overall, there were 75 cancer predisposition genes (Supplementary Table 3) covered in both studies. The germline mutation rate was significantly higher in the current Chinese lung cancer cohort than that in TCGA cohorts (94/1794 vs. 34/1017, $p = 0.026$, Chi-square test) (Supplementary Table 4). Similar trend was observed when the comparison focused only on LUAD (the dominate histologic type in the Chinese cohort), although the difference did not reach

statistical significance (66/1223 vs. 17/518, $p = 0.077$, Chi-square test). Twenty-nine of these 75 genes were identified to demonstrate P/LP germline mutations in at least one cohort (Supplementary Table 5). P/LP germline mutations in *BRCA2, FANCA, ATM, MUTYH, BLM, TP53, BRCA1, CHEK2, PMS2, NBN,* and *FANCC* were identified in both patient populations suggesting these genes may play important roles in genetic predisposition to lung cancers in both western and Chinese populations. Interestingly, *BRCA2* germline mutations were significantly more common in Chinese cohort than TCGA cohorts (14/1794 vs. 1/1017, $p = 0.015$, Fisher's exact test for all histologies; 11/1223 vs. 0/518, $p = 0.041$, Fisher's exact test for LUAD only) (Supplementary Table 5). In addition, 14 genes (*RAD51D, FANCD2, BRIP1, MSH6, PMS1, PALB2, RAD51C, SDHA, TSC2, BAP1, CDH1, FLCN, NF1,* and *RUNX1*) were exclusively identified in Chinese patients, while 4 genes (*RET, ERCC3, FANCG,* and *VHL*) were only detected in TCGA cohort. Among these genes, P/LP germline mutations in *RET* were significantly enriched in TCGA cohort (0/1794 vs. 3/1017, $p = 0.047$, Fisher's exact test) while no significant difference was found for other genes between Chinese cohort and TCGA cohort (Supplementary Table 5) likely due to the very low event rates in both cohorts.

**P/LP germline mutations in *BRCA1/2* and *TP53* may be associated with early onset of lung cancer.** Next, we sought to investigate whether the lung cancers with P/LP germline mutations have distinct clinical features compared to lung cancers without P/LP germline mutations. The germline mutation rate was not associated with gender, age, histology or stage (IV vs. I–III) either in univariate analysis or multivariate analysis (Supplementary Table 6). However, there was a trend that germline mutations were more common in younger patients, consistent with report in a Western lung cancer population[20]. The prevalence of P/LP germline mutation in patients under 40 was 8.57% vs. 5.29% in patients over 40 ($p = 0.218$, Chi-Square test)

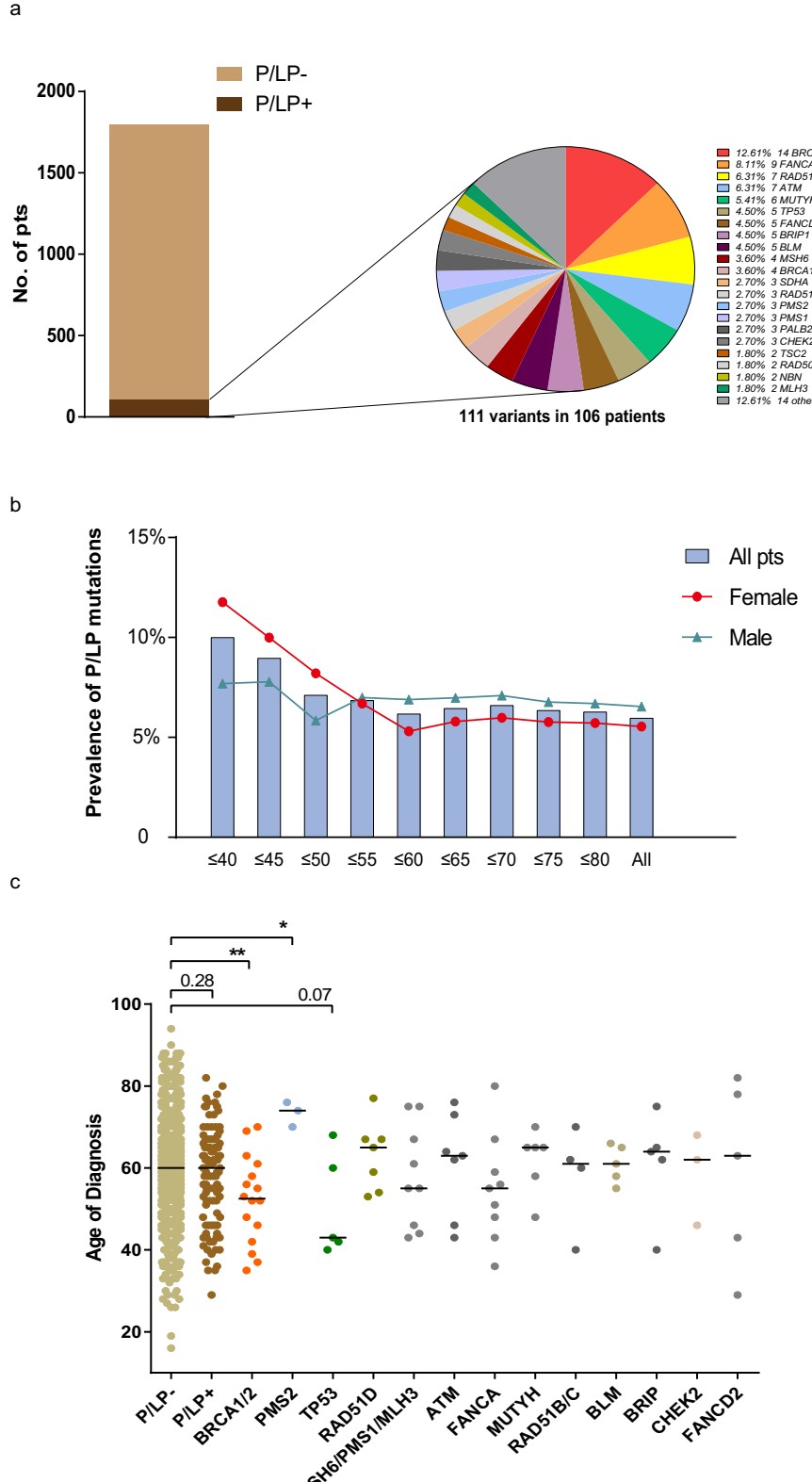

**Fig. 3 Distribution of P/LP germline mutations and the age at diagnosis. a** Bar plot indicated the prevalence of P/LP germline mutation (dark brown). The genes, number of patients, and mutation frequency of each gene are shown in the pie plot. Genes (*SLX4, RUNX1, RAD51B, RAD51, PTCH2, NF1, MRE11A, GALNT12, FLCN, FANCC, FAM175A, CDH1, BARD1* and *BAP1*) with germline mutations detected in one patient were grouped as others. **b** Frequency of pathogenic and likely pathogenic germline variants in patients of different ages ($n = 1715$ patients with information on age of onset). Bar plot and lines shows the frequency of germline variants in patients under certain age (bar) and frequency in female and male patients (lines). **c** The panels show the age of onset for patients without germline mutations ($n = 1611$ patients) (light brown dots) and patients with different germline genes ($n = 104$ patients) (dark brown dots). Horizontal lines indicate median age. *P* value is calculated by the Mann–Whitney test (*$p = 0.021$, **$p = 0.008$).

**Table 2 The enrichment of P/LP germline mutation in lung cancer cohort.**

| Genes | LC AC[a] | LC AN | LC AF | ChinaMAP AC[a] | ChinaMAP AN | AF | OR | 95% CI | p value |
|---|---|---|---|---|---|---|---|---|---|
| BRCA2* | 14 | 3574 | 0.0039 | 16 | 21160 | 0.0008 | 5.18 | 2.34–11.33 | 2.56E−05 |
| FANCA* | 9 | 3579 | 0.0025 | 4 | 21172 | 0.0002 | 13.30 | 3.71–59.17 | 1.1440E−05 |
| RAD51D* | 7 | 3581 | 0.0020 | 16 | 21160 | 0.0008 | 2.58 | 0.90–6.65 | 0.03921 |
| ATM* | 7 | 3581 | 0.0020 | 7 | 21169 | 0.0003 | 5.91 | 1.77–19.76 | 0.0018 |
| MUTYH | 6 | 3582 | 0.0017 | 41 | 20864 | 0.0019 | 0.86 | 0.30–2.05 | 1 |
| TP53* | 5 | 3583 | 0.0014 | 2 | 21174 | 9.445E−05 | 14.77 | 2.42–155.07 | 0.0010 |
| FANCD2* | 5 | 3583 | 0.0014 | 2 | 21174 | 9.445E−05 | 14.77 | 2.41–155.07 | 0.0010 |
| BRIP1* | 5 | 3583 | 0.0014 | 2 | 21174 | 9.445E−05 | 14.77 | 2.41–155.07 | 0.0010 |
| BLM | 5 | 3583 | 0.0014 | 11 | 21165 | 0.0005 | 2.68 | 0.73–8.39 | 0.0699 |
| BRCA1 | 4 | 3584 | 0.0011 | 12 | 21164 | 0.0006 | 1.97 | 0.46–6.50 | 0.2741 |
| MSH6* | 4 | 3584 | 0.0011 | 3 | 21173 | 0.0001 | 7.88 | 1.33–53.73 | 0.0107 |
| PMS2 | 3 | 3585 | 0.0008 | 7 | 21169 | 0.0003 | 2.53 | 0.42–11.09 | 0.1667 |
| PMS1* | 3 | 3585 | 0.0008 | 0 | 21176 | 0 | Inf | 2.44-Inf | 0.0030 |
| SDHA | 3 | 3585 | 0.0008 | 5 | 21171 | 0.0002 | 3.54 | 0.55–18.21 | 0.0968 |
| RAD51C* | 3 | 3585 | 0.0008 | 2 | 21174 | 9.445E−05 | 8.86 | 1.01–106.12 | 0.0242 |
| PALB2 | 3 | 3585 | 0.0008 | 6 | 21170 | 0.0003 | 2.95 | 0.48–13.83 | 0.1301 |
| CHEK2* | 3 | 3585 | 0.0008 | 1 | 21175 | 4.722E−05 | 17.72 | 1.42–924.85 | 0.0108 |

LC lung cancer, AC Allele counts (mutant allele count of P/LP mutation found from lung cancer cohort in certain genes), AN Allele number (Total allele count minus mutant AC in the cohort or database),
AF allele frequency, OR Odds Ratio, CI Confidence interval, Inf infinity.
*P value < 0.05 (calculated with Fisher's exact test).
[a]The same ACMG guideline and same criteria were applied to annotate P/LP mutations.

and the prevalence plateaued after 55 years old (Fig. 3b). This trend appeared to be primarily driven by patients with germline mutations in *BRCA1/2*, who were significantly younger than patients without germline mutations (median of 52.5 vs. 60 yrs, $p = 0.008$, Mann–Whitney test) or patients with other germline mutations (median, 52.5 vs. 62.5, $p = 0.016$, Mann–Whitney test) (Fig. 3c). In addition, there were five patients in our cohort who were identified to carry P/LP germline mutations in *TP53*. The median age of these five patients was 43 years old, younger than patients without germline mutations (60 yrs, $p = 0.07$, Mann–Whitney test) (Fig. 3c) or patients with other germline mutations (62.5 years old, $p = 0.16$, Mann–Whitney test) although the differences did not reach statistical significance likely due to small sample size. On the other hand, patients with P/LP germline *PMS2* mutations were older than patients without germline mutations (median, 74 vs. 60 yrs, $p = 0.021$, Mann–Whitney test), or patients with other P/LP germline mutations (median, 74 vs. 59.5 yrs, $p = 0.006$, Mann–Whitney test) (Fig. 3c).

**Somatic mutation landscape in non-small cell lung cancers with germline mutation.** We next investigated whether lung cancers with P/LP germline mutations have distinct somatic mutational landscape. To avoid false negative results ascribed to low tumor content in the specimens, 224 patients (including 187 with liquid biopsy samples and 37 with FFPE samples) without any somatic mutation detected were excluded for this analysis (Fig. 1). Furthermore, since 634 tumor specimens were formalin-fixed paraffin-embedded (FFPE) specimens, which are known to be associated with higher incidence of sequencing artifacts than fresh tissues, strict filtering criteria (see method for details) were applied for somatic mutation calls. DNA degradation increases with age of FFPE blocks, the quality of DNA heavily depends on storage time of FFPE specimens and the artifact rate is rather low if the FFPE blocks are <1 year old[21]. Fortunately, 551/634 (87%) FFPE specimens utilized in this study were collected within one year before DNA extraction. Nevertheless, we sought to determine the sequencing data quality before further analyses. Since FFPE sequencing artifacts usually present as low log odds (LOD) score (usually < 10), low VAF (usually < 10%), predominantly "C > T/G > A" transitions[22], we assessed the LOD scores and the

proportion of "C > T/G > A" transitions for all mutations included in this study. When particularly looking into these "high-risk" features, only 8 of 4784 (0.2%) mutations from FFPE specimens were C > T/G > A transitions with LOD score < 10 and VAF < 10%. Furthermore, the overall proportion of C > T transitions was similar for mutations identified from FFPE specimens compared to those from fresh tissue specimens (29.3% vs. 29.17%, $p = 0.951$, Chi-square test). Taken together, these data suggest that the impact of FFPE artifacts was minimal in this study.

Overall, the frequently mutated genes and tumor mutation burden (TMB, 5/Mb vs. 5/Mb, $p = 0.841$, Mann–Whitney test) were similar between patients with P/LP germline mutations and those without germline mutations (Fig. 4a). Interestingly, the group of patients with P/LP germline mutations were significantly enriched for somatic mutations in *MET* (7/92, 7.6% vs. 43/1434, 3.0%, $p = 0.027$, Fisher's exact test) mainly driven by *c-MET* exon 14 skipping mutations (4/92, 4.35% vs. 11/1434, 0.77%, $p = 0.010$, Fisher's exact test) (Supplementary Table 7) and *KRAS* mutations (15/92, 16.85% vs. 119/1434, 8.30%, $p = 0.015$, Chi-square test), while somatic mutations in *TP53* were significantly more common in patients without P/LP germline mutations (845/1434, 58.9% vs. 42/92, 45.7%, $p = 0.017$, Chi-square test) (Fig. 4b, Supplementary Table 8). These associations remained significant in multivariate analysis adjusting for gender, age and histology etc. with odds ratios of 2.127 (1.170–3.866 [95%CI], $p = 0.013$) for *KRAS* mutations; 5.536 (1.466–20.897 [95%CI], $p = 0.012$) for *c-MET* exon 14 skipping mutations and 0.577 (0.363–0.916 [95% CI], $p = 0.020$) for *TP53* mutations (Table 3). Otherwise, the prevalence of somatic mutations in other commonly mutated cancer genes in Asian lung cancer patients including *EGFR* (44.6% (41/92) in patients with germline mutations vs. 46.7% (675/1434) in patients without, $p = 0.720$) was comparable in patients with P/LP germline mutations and those without. These data suggested that although the common cancer gene mutations are similar between lung cancers with and without P/LP germline mutations, there might be genetic constraints in certain patients with cancer predisposition germline mutations.

**Germline mutations may have impact on mutagenesis of lung cancers.** We next sought to explore whether germline mutations could have impacted the mutagenesis in this cohort of lung

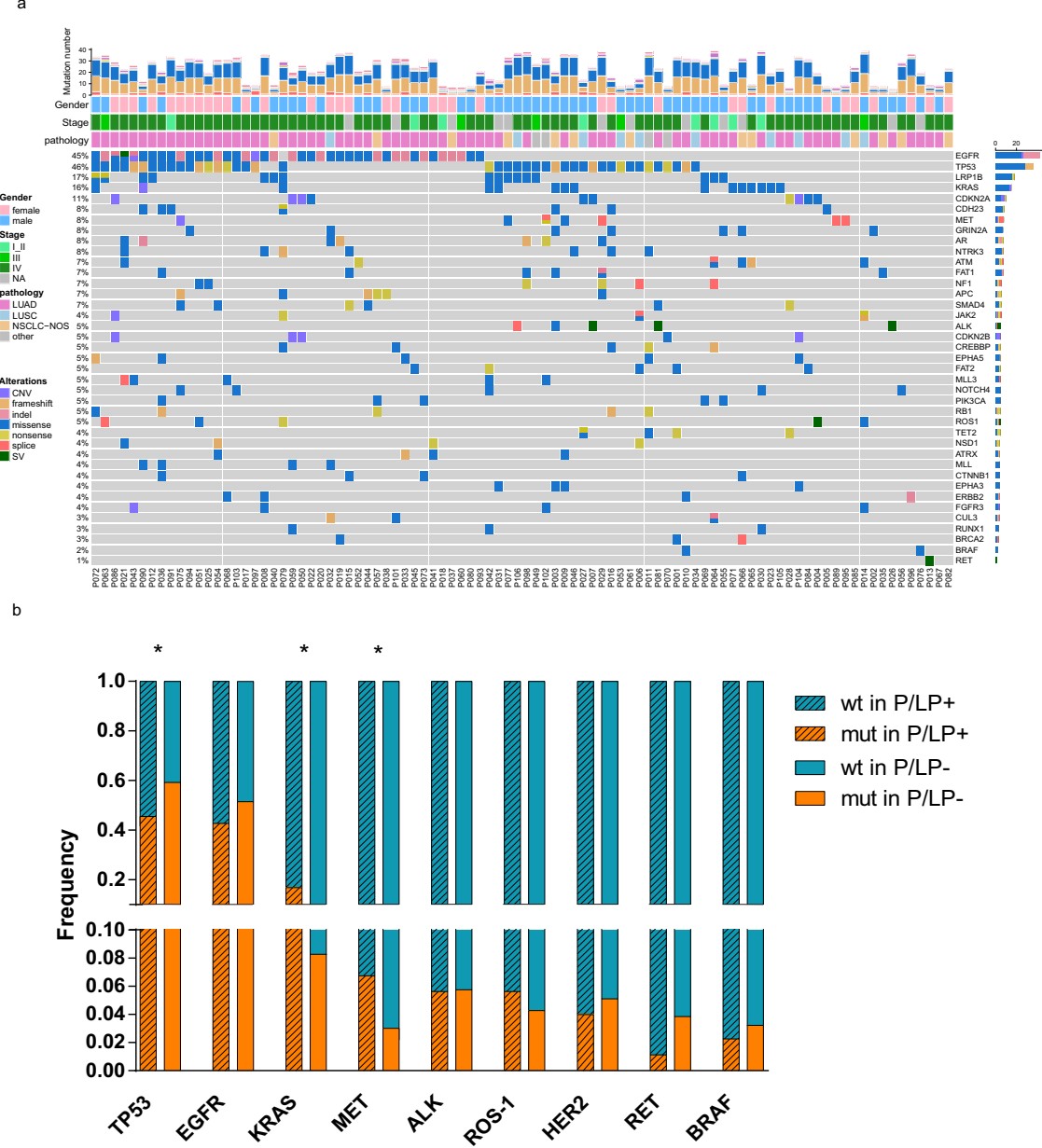

**Fig. 4 Commonly somatic cancer gene mutations in lung cancer patients with and without germline variants. a** Genomic landscape of somatic mutation in patients with P/LP germline mutation. Stacked plots (top) show the number of somatic mutations (SNV/indels, CNV and SV) in each tumor sample (column). The gender, histology subtype and clinical stage are shown on the bottom. Bar plot on the right shows the mutation frequency of each gene. **b** Normalized bar plot illustrates the frequency of nine commonly mutated genes. Stripped bars represent patients with P/LP germline variants. Open bars represent patients without pathogenic germline variants. *Statistic difference between two cohorts: *TP53*, p = 0.016. *KRAS*, p = 0.015 and *MET*, p = 0.027. P value is calculated with Chi-square test (*EGFR, TP53, KRAS*) or Fisher's exact test (*c-MET, ALK, ROS-1, ERBB2, RET, BRAF*).

cancers. Mutational Signature analysis can potentially indicate the contribution of certain genes, such as BRCA1/2 and MMR genes to tumorgenesis[23]. However, the numbers of mutations in each specimen from cancer gene panel sequencing applied were too small for reliable signature analysis. We therefore combined SNVs from all tumors with germline BRCA1/2 mutations (BRCA group) vs. tumors with germline MMR (MLH3, MSH6, PMS1, and PMS2) mutations (MMR group) respectively. With this caveat, we observed COSMIC mutational signature AC3 (associated with BRCA mutations) contributed to 33.5% of SNV in BRCA group vs. 20.1% in MMR group. On the contrary, the contribution of two signatures associated with MMR defect AC20 and AC26 accounted for 28.9% in SNVs in MMR group compared to 10.5% in BRCA group (Supplementary Fig. 1). These

data imply that germline mutations may contribute to the tumorgenesis by inducing related mutation type. However comprehensive studies with data at whole exome sequencing level are warranted to validate these findings.

## Discussion

Understanding genetic predisposition is critical for screening, prevention and treatment of patients with germline pathogenic alterations. Genome-wide analyses have offered new evidence on cancer pathogenic germline variants[10] and pan-cancer studies have described that deleterious germline mutations are present in many cancer types including lung cancers[8,9,20,24,25]. These pioneering studies have demonstrated that a considerable proportion

**Table 3 Correlation between germline mutation status and somatic mutations.**

| | Univariate analysis | | Multivariate analysis | |
|---|---|---|---|---|
| | OR (95%CI) | *p* value | OR (95%CI) | *p* value |
| | *KRAS* mutation | | | |
| Age of diagnosis | | 0.162 | | |
| Gender (Female vs. Male) | 0.228(0.141–0.368) | <0.001 | 0.229 (0.142–0.369) | <0.001 |
| Histology (LUSC vs. LUAD) | | 0.258 | | |
| Germline mutation (pos vs. neg) | 2.156(1.202–3.866) | 0.01 | 2.127 (1.170–3.866) | 0.013 |
| | *TP53* mutation | | | |
| Age of diagnosis | | 0.360 | | |
| Gender (Female vs. Male) | 0.626(0.509–0.769) | <0.001 | 0.678 (0.536–0.856) | 0.001 |
| Histology (LUSC vs. LUAD) | 2.153(1.477–3.138) | <0.001 | 1.895 (1.287–2.790) | 0.001 |
| Germline mutation (pos vs. neg) | 0.561(0.367–0.857) | 0.008 | 0.577 (0.363–0.916) | 0.020 |
| | *c-MET* 14 skipping | | | |
| Age of diagnosis | 1.071(1.017–1.127) | 0.009 | 1.076 (1.020–1.134) | 0.007 |
| Gender (Female vs. Male) | | 0.854 | | |
| Histology (LUSC vs. LUAD) | | 0.581 | | |
| Germline mutation (pos vs. neg) | 5.888(1.838–18.867) | 0.003 | 5.536 (1.466–20.897) | 0.012 |
| | *EGFR* mutation | | | |
| Age of diagnosis | 0.982(0.974–0.991) | <0.001 | 0.989 (0.978–0.999) | 0.038 |
| Gender (Female vs. Male) | 3.651(2.949–4.520) | <0.001 | 2.925 (2.289–3.737) | <0.001 |
| Histology (LUSC vs. LUAD) | 0.110(0.066–0.182) | <0.001 | 0.161 (0.096–0.271) | <0.001 |
| Germline mutation (pos vs. neg) | | 0.58 | | 0.967 |

Odds ratios, two-sided *p* value and 95% confidence intervals were calculated using logistic regression models. Factors with *p* value < 0.1 in univariate analysis were included for multivariate logistic regression models. Gene mutation counts of *TP53*, *KRAS*, and *EGFR* included SNV/indel, CNV and SV.
*CI* confidence interval, *OR* Odds ratio, *LUSC* lung squamous carcinoma, *LUAD* lung adenocarcinoma.

of cancer patients may carry genetic predisposition germline mutations advocating for further studies to address this important question. However, these pioneer studies were conducted almost exclusively in western population. The germline mutation landscape in Asian lung cancer patients has not been systematically investigated. A few studies on this effort have focused on particular genes or pathways such as a large cohort of 12,833 Chinese lung cancer patients from Dr. Lu's team on the germline *EGFR* mutations[26] and a study led by Dr. Sun et al. on the germline mutations of MMR genes (*MLH1*, *MSH2*, *MSH6*, and *PMS2*) in a cohort of 1179 Chinese lung cancer patients[27]. Our study provided the first set of data on the clinical and genomic features of Chinese lung cancer patients with P/LP germline predisposition mutations using a relatively large gene panel and revealed that a substantial proportion (5.91%) of Chinese lung cancer patients could carry P/LP germline mutations. Considering the large cardinal number of lung cancer patients in China, these carriers represent a very large patient population.

As expected, certain genetic predisposition genes were shared between Chinese lung cancer patients and western lung cancer patients from TCGA. However, the prevalence of P/LP germline mutations was different in these two lung cancer patient cohorts. Some genetic predisposition genes were unique to the Chinese or the TCGA lung cancer cohort implying a potential difference in genetic influences and/or exposures between these two patient populations. One caveat is that compared to TCGA cohorts, patients in the current Chinese cohort were younger, with more female patients and stage IV diseases (Supplementary Table 9). These important differences could have potentially confounded the observed higher incidence of P/LP germline mutations in the Chinese cohort. However, in the current cohort, incidence of P/LP mutations did not seem to correlate with gender, age, histology or stage (Supplementary Table 6). Similarly, in a study on western lung cancer cohort, incidence of P/LP genetic mutations was not different between histologies[20]. Taken together, these data implied that distinct ethnic background and possibly

exposure history may be the main reasons for the observed differences in germline mutations between Chinese and Western lung cancer patients.

With the modest sample size fully acknowledged, we attempted to address the question whether lung cancer patients carrying genetic disposition germline mutations have unique clinical and molecular features. Of particular interest, the age of onset was significantly younger in patients with germline mutations in *BRCA1/2*. These observations were consistent with a previous pan-cancer analysis[8] and studies on hereditary breast cancer and colorectal cancer[28,29]. Similarly, patients with P/LP germline mutations in *TP53* also appeared to be younger than patients without P/LP germline mutations or patients with other germline mutations although the difference did not reach statistical difference likely due to small sample size (Fig. 3c). This is in line with previous findings that germline mutations in *TP53* are associated with early onset of various cancers in patients with Li-Fraumeni syndrome[30,31]. These data, if validated, advocate for screening of lung cancer at younger age in individuals with certain cancer predisposition germline mutations.

Another interesting finding was the lack of association between P/LP germline mutations and *EGFR* mutations, which are well documented to be far more prevalent in Asian lung cancer patients than western populations. The exact mechanisms remain unknown, but different genetic background in different ethnic populations is thought to be one of the potential reasons. Germline *EGFR* mutations such as T790M have been reported in hereditary lung cancers[32]. In above mentioned study on 12,833 Chinese lung cancer patients, germline *EGFR* mutations were identified in 14 patients (0.11%)[26]. Interestingly, germline *EGFR* T790M mutation was identified in only 1 of 5675 (0.02%) Chinese lung cancer patients carrying somatic *EGFR* mutations, much lower than 1–4% in *EGFR*-mutant Caucasian lung cancer patients[33,34] further highlighting the potential ethnic and etiologic differences between Chinese and western patient populations. In the current study, we did not detect any *EGFR* germline

mutations in the 1794 lung cancer patients. Furthermore, the prevalence of somatic *EGFR* mutations did not appear to associate with P/LP germline mutations. Taken together, these data suggest the genetic basis for germline *EGFR* mutations and somatic *EGFR* mutations in Chinese lung cancer patients is different and P/LP germline mutations unlikely account for predisposition of somatic mutations in *EGFR* in Chinese lung cancer patients.

One major caveat is that in many studies including the current study, P/LP germline mutations were annotated according to the ACMG guidelines, which were mainly based on the data and experience from Caucasian patients. Because of the distinct genetic background, these guidelines may not always apply to other ethnic populations. In the current study, for example, *MUTYH*, a gene encoding a DNA glycosylase involved in oxidative DNA damage repair and associated with heritable predisposition to various cancers, particularly colorectal cancer[35,36], demonstrated similar AF between lung cancer patients and non-cancer individuals suggesting that the germline mutations in *MUTYH* were not associated with lung cancer risk. It is worth noting that an *MUTYH* variant c.934−2 A > G (rs77542170) is defined as a "P/LP" mutation based on ACMG guidelines. Indeed, the AF of *MUTYH* c.934−2 A > G in non-cancer individuals from the Genome Aggregation Database (GAD) was only 0.11% (allele count (AC): 312 of 28,2820) in line with this annotation. However, 307 of the 312 AC were from the East Asians (EAS) with an AF of 1.5% (307/19952) in EAS, 13.6 times higher than that of the whole GAD population ($P = 2.2E−16$). In addition, there were five EAS individuals harboring homozygous alleles of this variant (gnomAD, https://gnomad.broadinstitute.org). Moreover, two studies on Japanese patients reported that AF of this *MUTYH* c.934-2 A > G mutation in gastric cancer patients[37] and colorectal cancer patients[38] was no different compared to non-cancer individuals. These results suggested that *MUTYH* c.934-2 A > G (rs77542170) most likely is not a pathologic germline mutation for EAS individuals. These findings highlighted the profound impact of ethnicity on defining P/LP germline mutations and emphasized the importance of taking ethnicity into consideration when annotating P/LP germline mutations. Furthermore, because we only included P/LP germline mutations based on ACMG guidelines, there may be other cancer predisposition genes or mutations unique to Asian patients that we were not able to identify in the current study. There have been efforts to fill this void. For example, the China food and drug institute established a standard database based on the interpretation of genetic variation in Chinese population with the goal to establish a reference system for performance evaluation of BRCA genetic testing[39]. Future studies on large cohort of Asian cancer patients using more comprehensive panel, ideally at exome level are warranted to establish clinical germline database for Asian cancer patients.

The majority of studies on genetic predisposition are primarily based on the association between the presence of certain germline mutations and cancer incidences. In our study, we found that the frequencies of P/LP germline mutations were significantly higher in lung cancer patients than the 10,588 non-cancer Chinese individuals (Table 2). In addition, evidence of second-hit of genes with P/LP germline mutations was found in 18.6% of tumors. These results indicated the connection between these germline mutations and lung carcinogenesis. Other bioinformatics approaches, such as mutational signature analysis[40], homologous recombination deficiency score[41] analysis etc. could potentially provide further support to the contribution of these germline mutations to lung cancer development. Unfortunately, our data was limited by small numbers of mutations from panel sequencing for such analyses. Nevertheless, these association-based studies have served as the bases for establishing guidelines such as ACMG, which are of value to determine strategies for screening and prevention of certain cancers. However, from cancer biology standpoint, presence of a mutation does not necessarily mean it is causative. Functional studies including genetic animal models are eventually needed to determine the impact of certain germline mutations on carcinogenesis.

One inherent limitation of our study, as a retrospective real-world data mining study, is that clinical information including smoking history, treatment response and survival data etc. were not available from many patients, which precluded us being able to explore some very important questions such as the impact of P/LP germline mutations on mutational signatures, treatment response and prognosis. Nevertheless, as the first study, our data suggested that substantial proportion of Chinese lung cancer patients may carry germline mutations. These patients with germline mutations may have distinct clinical and molecular features and the genes accounting for lung cancer predisposition in Asian patients may be different from those in western populations. These results highlighted again the need for future prospective studies on larger cohorts of Asian patients to identify cancer disposition genes unique to Asian populations as well as to define the clinical and genomic features of cancer patients with germline mutations for precise cancer prevention, screening and treatment.

## Methods

**Patient cohort and samples**. Cohort in this study encompassed 1794 lung cancer patients (Supplementary data 2), who were subjected to target capture NGS of 1021 cancer genes in tumor DNA and paired germline DNA as part of the clinical care. The study was approved by the Ethics Committee of Hunan Cancer Hospital and all participants signed a written informed consent.

**Sample processing, DNA extraction and Quantification**. Liquid biopsy samples (peripheral blood, ascitic effusion, pleural effusion, pericardial effusion and cerebrospinal liquid) were collected in Streck vacutainer tubes (Omaha, NE) and processed within 48 h to separate the supernatant by centrifugation at 1600 *g* for 10 min. Buffy coat from peripheral blood was kept for DNA extraction as germline control. The supernatant was transferred to microcentrifuge tubes, and further centrifuged at 16,000 *g* for 10 min to remove remaining cell debris. Separated liquid biopsy samples and buffy coat were stored at −80 °C until DNA extraction. Separated liquid biopsy samples were isolated for cell free DNA (cfDNA) using a QIAamp Circulating Nucleic Acid Kit (Qiagen, Hilden, Germany). Buffy coat DNA and FFPE tumor tissue DNA were extracted using the DNeasy Blood & Tissue Kit (Qiagen). DNA concentration was measured using Qubit fluorometer 3.0 (Life Technologies) and the Qubit dsDNA HS (High Sensitivity) Assay Kit (Invitrogen, Carlsbad, CA, USA). The cfDNA size distribution was evaluated using an Agilent 2100 BioAnalyzer and a DNA HS kit (Agilent Technologies, Santa Clara, CA, USA). The sample quality was assessed based on the following criteria: total amount ≥30 ng for cfDNA samples or ≥100 ng for tumor FFPE samples; fragment length for cfDNA samples was distributed with a dominant peak at 170 bp proximately[42,43].

**Library construction, target enrichment and sequencing**. Before library construction, DNA from buffy coat peripheral blood lymphocytes (PBL) or from FFPE samples was sheared to 200–300 bp fragments using a Covaris S2 ultrasonicator (Covaris, Woburn, MA, USA). Indexed Illumina next-generation sequencing (NGS) libraries were prepared from PBL DNA, tumor DNA, and liquid biopsy DNA using the KAPA Library Preparation Kit (Kapa Biosystems, Wilmington, MA, USA). The region of frequently mutated 1021 genes (Supplementary data 3) in solid tumors were enriched using a custom SeqCap EZ Library (Integrated DNA Technology, Coralville, IA, USA). Captured hybridization was performed using the manufacturer's protocol. Following hybrid selection, the captured DNA fragments were amplified and then pooled to generate several multiplex libraries. Of note, for liquid biopsy samples, the duplex sequencing based on a unique identifier tag (UID) were applied to filter repeatedly errors in the consensus bidirectionally and rectify sequencing errors mostly introduced by PCR/sequencing and modify the base quality. Finally, the libraries were performed on NovaSeq6000 or Hiseq3000 Sequencing System (Illumina, San Diego, CA) with 2 × 101 bp paired-end reads. The TruSeq PE Cluster Generation Kit V3 and TruSeq SBS Kit V3 (Illumina, San Diego, CA, USA) were used according to the manufacturer's recommendations[44].

**Germline variant analysis and annotation**. Base-calling was conducted through the Illumina analysis pipeline (CASAVA 1.8). Low-quality data was removed and

each barcoded dataset was separated. Burrows–Wheeler Aligner was used to map reads to the reference genome GRCh37/hg19. GATK (Version 3.6)[45] (haplotype caller in single-sample mode with duplicate and unmapped reads removed using defaulted parameters) was used to detect single-nucleotide variants (SNVs) and small insertions and deletions (indels) from germline DNA samples extracted from blood. Variants in 94 genes (selected from Genetic Testing Registry[46] (GTR, www.ncbi.nlm.nih.gov/gtr/) and NCCN Genetic/Familial High-Risk Assessment guidelines[46,47]) (Supplementary Table 10) were included for further annotation. The following filtering criteria were applied. (1) A minimal mapping quality of 25 was used to ensure high quality reads. (2) Only germline mutations that meet the following criteria were included: A. Sequencing depth at the targets >50× (78×–510×, mean 265× for the data used in this study); and B. Variant allele frequency (VAF) > 25% (25%–55%, mean 47.3% for the data used in this study). (3) All germline mutations were manually verified using IGV browser. Common SNPs in ≥1% of population in the 1000 genomes, ExAC and ExAC Asian databases were filtered out. Variants were matched with those in the ClinVar, HGMD or an inhouse database, and then were manually confirmed and annotated as pathogenic, likely pathogenic, uncertain significance, likely benign or benign according to 2015 ACMG Guideline[16]. Various types of evidence classified as PVS1 (pathogenic very strong 1), PS2 (pathogenic strong 2), PS3 (pathogenic strong 3), PM6 (pathogenic moderate 6) and BS3 (benign strong 3) in ACMG guideline were confirmed according to the recommendation of application ACMG guideline[48–50] by Clinical Genome Resource (Clingen) (https://www.clinicalgenome.org/).

**Genetic architecture of Chinese lung cancer patients**. SNP data from 1000 genomes project phase 3 ($n = 2054$) was utilized. Following criteria were employed to select the SNPs covered in this 1021 cancer gene panel: minor allele frequency ≥ 1% (common and low-frequency variants), genotyping rate ≥ 90%, Hardy–Weinberg–Equilibrium $P > 0.000001$, and removing one SNP from each pair with r2 ≥ 0.5 (in windows of 50 SNPs with steps of 5 SNPs). The mean pairwise Fst differences between the Chinese lung cancer patients and different ethnic populations in the 1000 genome population (1KGP) were calculated using EIGENSOFT (Version 7.2.1). Principal component analysis (PCA) was performed using autosomal bi-allelic SNPs. The PCA was performed with the final SNPs using PLINK[51] (Version 1.9) and EIGENSOFT[52,53] (Version 7.2.1).

**Somatic sequencing data analysis**. A minimal mapping quality of 25 was required to ensure high-quality somatic reads. Somatic SNVs in tumor DNA were called using MuTect (Version 1.4) and NChot[54]. GATK (Version 3.6)[45] was used to identify indels. (Supplementary data 4). Somatic mutations that meet the following criteria were included for further analyses: sequencing average depth >100× in germline DNA, >500× in tumor DNA (1000 in ctDNA), minimal VAF > 1% in tumor DNA (0.5% in ctDNA), the ratio of AF in case/control (tumor /germline) >3 and at least 4 supportive reads (both tissue and ctDNA). For hotspot mutations (EGFR 19del, EGFR L858R, EGFR T790M, KRAS G12, MET 14 exon skipping, BRAF V600E etc.), the requirements were: the sequencing depth >20, 3 (for SNV) or 2 (for indel) supportive reads, and the ratio of AF in case/control (tumor /germline) >3. In addition, all mutations were manually verified with IGV browser. Somatic copy-number variation (CNV) were identified with CONTRA (Version 2.0.8) and calculated as the ratio of adjusted depth between tumor DNA and germline DNA (Supplementary data 5). Loss of heterozygosity (LOH) was analyzed with Facets (Version 1.0.1)[55]. For structural variations (SV) (Supplementary data 6), probes were designed to capture selected exons and introns of RET, ALK, ROS1, and NTRK1 oncogenes based on previously reported SVs. An in-house algorithm was used to identify split-read and discordant read-pair. In addition, mutations associated with clonal hematopoiesis were filtered out[56]. R package "YAPSA" was applied to deconstruct signatures from combined SNV from each groups. Signatures (AC) contributed by over 3% of SNVs were displayed.

**Calculation of tumor mutation burden**. The tumor mutation burden (TMB) was calculated as the number of non-silent somatic mutations (non-synonymous SNV, indel and splice ± 2) per mega-base (1 Mb) of coding genomic regions sequenced (1 Mb for this 1021 panel). To avoid the false negative results that may confound the TMB calculation, only samples carrying at least one mutation with VAF > 0.03 (in tissue sample) or >0.005 (in ctDNA sample) were included. Other specimens were classified as TMB-unevaluable. CNV or SV was not included for TMB calculation.

**Statistical analysis**. Mann–Whitney test was employed to compare age and TMB between groups. The Chi-square test or Fisher's exact test was performed to test frequency between groups. All statistical analysis was performed with SPSS (v.23.0; STATA, College Station, TX, USA) or GraphPad Prism (v. 6.0; GraphPad Software, La Jolla, CA, USA) software. Statistical significance was defined as a two-sided $p$ value of <0.05.

**Reporting summary**. Further information on research design is available in the Nature Research Reporting Summary linked to this article.

## Data availability
Patient deidentified clinical and mutation data (both germline and somatic mutations) were provided in the Supplementary data 1–6. The Fastq data of all samples were deposited in the GSA-Human (Genome Sequence Archive for Human in BIG Data Center, Beijing Institute of Genomics, Chinese Academy of Sciences, http://gsa.big.ac.cn/gsa-human, https://ngdc.cncb.ac.cn/gsa-human/browse/HRA001610). The data are available under controlled access and may be requested by completing the application form via GSA-Human System. Data acquisition is granted by the corresponding Data Access Committee. The approximate response time for accession requests is about 2 weeks. Additional guidance are shown on the GSA-Human System website [https://ngdc.cncb.ac.cn/gsa-human/document/GSA-Human_Request_Guide_for_Users_us.pdf]. The reference genome used in this study was GRCh37/hg19. SNP data from 1000 genomes project phase 3 were used for SNP architecture analysis (https://www.internationalgenome.org/data-portal/data-collection/phase-3).
Germline mutation data form the Cancer Genome Atlas (https://www.cell.com/cell/fulltext/S0092-8674(18)30363-5) were used to compare the landscape of germline mutations and jung carcinogenesis (https://www.nature.com/articles/s41422-020-0322-9), all variants could be accessed through the ChinaMAP browser (www.mBiobank.com). A complete list of germline mutation can be found in Supplementary Data 1. Epidemiological information can be found in Supplementary Data 2. Gene list and region in sequencing panel can be found in Supplementary Data 3. somatic variation, copy-number variation and structural variation can be found in Supplementary Data 4–6.

## Code availability
The code used to identify split-read and discordant read-pair is publicly available at https://github.com/GenePlus/ncsv. R script and SNP data utilized to analyze to genetic architecture are deposited on git-hub (https://github.com/geneplus-beijing/1000GenomesProject.git). R script to deconstruct signatures from combined SNV from each group are available at https://github.com/geneplus-beijing/Signature.git). The R scripts to reproduce the analyses and plots reported in this paper are available from the corresponding authors upon request.

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

## Acknowledgements

Special thanks to Prof. Jianjun Zhang for his contribution to guiding the data analysis, helping with writing this paper. We thank Prof. Kai He, Prof. David P, Carbone, Prof. John V. Heymach, Prof. Ignacio I. Wistuba, Prof. P. Andrew Futreal for their help with improving the paper. We sincerely thank Dr. Xuan Gao, Dr. Zongbi Yi, Dr. Guofeng Zhao, Dr. Dongqiang Zeng Dr. Shengjie Lin and Dr. Chengxing Xia for their constructive discussions. We thank the patients, families, nurses and investigators who participated in this study. This work was supported by Cancer Foundation of China (Grant number LC2016W09), Changsha Science and Technology Bureau Foundation (Grant number kq1706039), Hunan Health Commission Foundation (Grant number B2019090), Wu Jieping Medical Foundation (Grant number 320.6750.19088-11) and Chinese Society of Clinical Oncology (CSCO) Research Foundation (Grant number Y-2019Genecast-024), Beijing Medical Health Public Welfare Foundation. (YWJKJJHKYJJ-B7452, Hunan Cancer Hospital Climb plan (ZX2020005-5) and Hunan Provincial Natural Science Foundation of China (2021JJ30430) to L.W., National Natural Science Foundation of China (Grant number 81972195) and Hunan Provincial Key Area R&D Program Project (Grant number 2019SK2253) to F.Y., Natural Science Basic Research Program of Shaanxi (2020JC-01) to J.W.

## Author contributions

L.W. conceived the overall study, X.X and L.C designed the study, Y.G. constructed and managed the sequencing and variant-calling pipelines. N.W., X.P., M.J., Jia.L., R.Z., F.X., L.X., B.C., B.L., and F.Y. collected the data. R.C. and Y.Y. produced the data and helped compile the clinical data. P.D., C.C., and J.H. did the variants annotation and manually check. W.P., Jin L., and J.B. analyzed the data, Y.Z. aided in statistical analysis. X.X., L.W. supervised the research. W.P., Jin.L. wrote the paper, L.W., L.Y., X.Y., X.X., and J.W. participated in optimization of the paper. All authors contributed to writing the paper. All authors reviewed and approved the final draft. L.W., and F.Y. jointly supervised this work.

## Competing interests

L.W. reports personal fees from AstraZeneca, Roche, Bristol-Myers Squibb, MSD, Pfizer, Lilly, Boehringer Ingelheim, Merck, Innovent and Hengrui outside the submitted work. J.B., Y.Y., R.C., Y.Z.,C.C., Jin L., J.H., P.D., Y.G., L.C., L.Y., and X.Y are current employees of Geneplus-Beijing. The remaining authors declare no competing interests.
