## [Peer Review File · Nature Communications]

Clinical and genomic features of Chinese lung cancer patients with germline mutationsReviewers' comments:

Reviewer #1 (Remarks to the Author):

Summary: The overarching aim of this work is to define the germline mutational landscape of Chinese lung cancer patients. Previous work has shown some evidence that germline mutations predispose individuals to lung cancer, however this work has been done predominantly in Caucasian populations. The authors also highlight that, identifying such predisposition genes is often crucial in determining therapeutic strategies and in identifying at-risk individuals. To address this paucity in information specific to the Chinese lung cancer population the authors sequence germline DNA from 1794 Chinese lung cancer patients targeted towards 1021 cancer genes. 35 of these genes harbor pathogenic/likely pathogenic (P/LP) mutations. To evaluate differences between Western/(predominantly)Caucasian populations and Chinese populations authors compare data from the TCGA cohort. 29 cancer disposition genes were present in both cohorts. And finally, the authors assessed the prevalence of co-occurring somatic mutations with germline mutations.

The main conclusions of the paper are that 5% of the Chinese lung cancer patients have pathogenic/likely pathogenic variants in known cancer genes. BRCA1/2 mutations are associated with earlier onset. A weaker conclusion (limited by statistical power) is the different prevalence of some of these mutations in Chinese vs. TCGA cohorts.

Overall the paper contributes new and valuable information to the lung cancer community. This value would be enhanced by making the somatic variants publicly available in a data portal or other file, and by providing additional information on variant calling strategy.

Comments/Questions/Clarifications:

1. While calling mutations what filters, if any did the authors employ to account for different sources of DNA, including FFPE which is known to introduce artifacts.
2. How will the data be shared? Please provide a dbGAP or other accession number. Alternatively, data could be uploaded to cBioPortal.org
3. In Figure 1, authors mention that there are 224 cases with no somatic mutations detected. Could the authors provide a possible explanation as to why they do not detect any somatic mutations in these patients?
4. Did the authors look at any other sequenced cohorts that includes Chinese populations at germline mutations to corroborate their findings?
5. Was any validation performed on the identified variants?
6. How did the authors calculate tumor mutational burden?
7. Fig. 3a – “Alternations” should be “alterations”
8. The authors could also provide some commentary on TP53 mutations and earlier age of diagnosis? Was the difference in age of onset significant for TP53 mutations? Are there specific TP53 variants that predict earlier onset more than others?
9. One obvious caveat of this study is the ACMG classification that was used to determine the pathogenic potential of germline mutations. Authors also have identified this. Can the authors offer a potential strategy to address this in the future?
10. In supplementary table 1 the title mentions 107 patients detected with P/LP mutations, however in the text mentions 106? Correction needed?
11. There were a number of minor typos throughout the text that should be corrected before publication.

Reviewer #2 (Remarks to the Author):

This is the first large scale analysis of germline mutation in lung cancer of Asian. The results are interesting, however, the reviewer feels that more data are needed to understand roles of observed germline mutations.

1. It is unclear whether germline mutations in BRCA1/2 and other homologous recombination-related genes contributed to tumorigenesis or not. Comparison of HRD (homologous recombination deficiency) scores and/or Cosmic signature 3 fractions between P/LP(+) and P/LP(-) populations are needed to draw a conclusion.
2. It is unclear whether the germline mutations in mismatch repair genes contributed to tumorigenesis or not. Comparison of tumor mutation burden as well as Cosmic signature 6 fractions between P/LP(+) and P/LP(-) populations are needed to draw a conclusion.
3. It is unclear whether the germline mutation frequencies in lung cancer patients are higher than those of non-lung cancer population? So, etiologically, the results do not tell a lot for lung carcinogenesis.
4. How the authors declare that the germline mutation-positive patients are genetically ordinal Chinese? Is it possible to perform principle component analysis using SNP information obtained by NGS analysis?

Point-by-point response

Reviewer #1 (Remarks to the Author):

Overall the paper contributes new and valuable information to the lung cancer community. This value would be enhanced by making the somatic variants publicly available in a data portal or other file, and by providing additional information on variant calling strategy.

Author's response: We appreciate reviewer's favorable comments and constructive suggestions.

1. While calling mutations what filters, if any did the authors employ to account for different sources of DNA, including FFPE which is known to introduce artifacts.

Author's response: We appreciate this critical question and agree with the reviewer that the data quality is essential, particularly when FFPE specimens were utilized. Therefore, strict quality control including various filtering strategies was applied before any further analyses.

The sequencing was done in a clinical laboratory that has been stringently examined and certified by Next-Generation Sequencing Solid Tumor (NGSST) of College of American Pathologists-Proficiency Testing (CAP-PT) and China National Center for Clinical Laboratories (NCCL). The same strict bioinformatics process was applied for all data utilized in this study, as for clinical report.

For the germline mutation calling (DNA from peripheral blood for all patients), the following filtering criteria were applied.

- 1) A minimal mapping quality of 25 was used to ensure high quality reads.
- 2) Only germline mutations that meet the following criteria were included: A. Sequencing depth at the targets $>50\times$ ($78\times$ - $510\times$, mean $265\times$ for the data used in this study); AND B. Variant allele frequency (VAF) $>25\%$ (25% - 55% , mean 47.3% for the data used in this study).
- 3) All germline mutations were manually verified using IGV browser.

For the somatic mutation calling, the following processes and filtering criteria were applied.

- 1) A minimal mapping quality of 25 was used to ensure high quality reads.
- 2) Only somatic mutations that meet the following criteria were included: A. Sequencing depth >100 in germline DNA + >500 in tumor DNA (or >1000 in ctDNA); AND B. VAF $> 1\%$ in tumor DNA (or $> 0.5\%$ in ctDNA); AND C. > 4 supportive reads (for both tumor DNA and ctDNA); AND D. The VAF ratio (tumor/germline) > 3 .
- 3) For hotspot mutations (EGFR 19del, EGFR L858R, EGFR T790M, KRAS G12, MET 14 exon skipping, BRAF V600E etc.), the filtering criteria were: A. the sequencing depth >20 ; B. VAF $> 1\%$ in tumor DNA (or $> 0.5\%$ in ctDNA); AND C. > 3 (for SNV) or 2 (for indel) supportive reads; AND D. The VAF ratio (tumor /germline) > 3 .
- 4) All mutations were manually verified with IGV browser.^[1,2]

We agree with the reviewer that these technical details are critical for readers to determine the quality of the data. We therefore added these filtering criteria to the method section in the revised manuscript (Page 12, line 7-13, Page 12, line 40-page 13 line 7).

As the reviewer pointed out, FFPE specimens are associated with higher rate of sequencing artifacts.

As DNA degradation increases with age of FFPE blocks, the quality of DNA from FFPE tissues heavily depends on storage time. The artifact rate is rather low if the FFPE blocks are less than 1 year old^[3-5]. Fortunately, the FFPE specimens used in this study were relatively “fresh” with 551/634 (87%) FFPE specimens collected within one year before DNA extraction. Nevertheless, we sought to determine the sequencing data quality. Since FFPE sequencing artifacts usually present as non-recurrent, low log odds (LOD) score (usually <10), low VAF (usually <10%), predominantly “C>T/G>A” transitions^[6], we looked into the LOD scores and the proportion of “C>T/G>A” transitions for all mutations included in this study. We first particularly looked into mutations with these “high-risk” features and only 8 of 4,784 (0.2%) mutations from FFPE specimens were C>T/G>A transitions with LOD score < 10 and VAF < 10%. Furthermore, the overall proportion of C>T transitions was similar for mutations identified from FFPE specimens compared to those from fresh tissue specimens (29.3% versus 29.17%, $p=0.951$, Chi square test) (**Reviewer Figure 1**). Taken together, these data suggest that the impact of FFPE artifacts would be minimal in this study. We agree with the reviewer that the quality control is important for sequencing data analysis from FFPE specimens and the information may be helpful for the field. Therefore, we added a paragraph on the quality control in the results section (**Page 7, line 10 - line 28**).

Reviewer Figure 1. The proportion of C>T transitions in FFPE specimens versus fresh specimens. The proportion of C>T/G>A transitions (blue) in FFPE specimens (29.3%, 1389/4784) versus fresh specimens (29.17%, 367/1258) ($p = 0.951$, Chi square test).

2. How will the data be shared? Please provide a dbGAP or other accession number. Alternatively, data could be uploaded to cBioPortal.org.

Author’s response: We thank the reviewer for this question and we agree that sharing the data is an important aspect of contribution to the academic community. As such, we have deposited relevant data to China National Gene Bank (CNGBdb) according to local laws and regulations. The data that support the findings of this study is now available at CNGBdb (<https://db.cngb.org/cnsa/>, accession number CNP0001060). Qualified researcher can apply for controlled access to the data. We updated data availability in the revised manuscript (**Page 13 line 33 - line 36**).

3. In Figure 1, authors mention that there are 224 cases with no somatic mutations detected. Could the authors provide a possible explanation as to why they do not detect any somatic mutations in these patients?

Author's response: We sincerely thank reviewer for this critical question. As discussed in our manuscript, one caveat for this retrospective real-world study was the heterogeneous nature of the tumor specimens. Among the 224 NSCLC specimens without somatic mutation detected, 185(82.2%) were ctDNA specimens. When specimen type was considered, 41 of 944 (4.3%) tissue DNA samples were negative for any mutations, while 183 of 806 (22.7%) ctDNA samples were negative. Therefore, some of these may be false negative due to low concentration of tumor DNA in circulation. Since the main focus of this study was germline variation, these patients without somatic mutations detected were included for the germline mutation analyses. However, to avoid false negatives that could confound the analysis, for the correlation between germline mutations and somatic mutational landscape, all mutation-negative specimens were excluded.

4. Did the authors look at any other sequenced cohorts that includes Chinese populations at germline mutations to corroborate their findings?

Author's response: We thank the reviewer for this constructive advice. To the best of our knowledge, very few Chinese patients were included in previous major studies on the germline mutations. There were a few studies focusing on certain genes or smaller gene panel. For example, Lu's et al^[7] studied the germline mutations in EGFR in Chinese population and identified only 1 *EGFR* T790M mutation, similar to the finding in our cohort (1/12833 vs. 0/1794, $p=1$, Fisher's exact test). Another study focused on germline mutations in mismatch repair (MMR) genes MLH1, MSH2, MSH6, and PMS2 in a cohort of 1,179 Chinese lung cancer patients^[8], in which pathogenic or likely-pathogenic germline mutations of MMR genes were detected in 6 (0.5%) patients. The median age at diagnosis of these cases was 68.5 years old. These findings were similar to our study. We updated the discussion of these studies in revised manuscript (Page 8 line 19 - line 25).

5. Was any validation performed on the identified variants?

Author's response: We thank the reviewer for this critical question and we agree with the reviewer on the importance of validity of NGS data. Since our assay is a clinical assay that have been extensively utilized and validated during R&D and subsequent clinical use, we initially did not perform independent validation for these data. To address the reviewer's request, we made extensive effort to obtain DNA to validate the data used in this study and were able to find two specimens (P001 positive for BRCA1 c.1961dupA (p.Y655Vfs*18) and P020 for BRCA2 c.6031T[2] (p.S2012Qfs*5)) for Sanger sequencing. As shown in **Reviewer Figure 2**, **Reviewer Figure 3**, both mutations were validated. Furthermore, we searched a pan-cancer germline mutation database at Geneplus-Beijing Institute and identified 39 additional specimens (from patients with other cancers) with P/LP mutations defined by exactly the same criteria in this study and all 39 germline mutations were successfully validated by Sanger sequencing.

Reviewer Figure 2. Sanger validation of BRCA1 c.1961dupA (p.Y655Vfs*18)

Reviewer Figure 3. Sanger validation of BRCA2 c.6031T[2] (p.S2012Qfs*5).

In addition, following are some previous validation efforts as your reference.

- 1) In 2015, Geneplus-Beijing Institute developed NGS-based genetic testing platform to detect mutations for common hereditary cancers^[9]. Single base substitutions across 115 hereditary cancer related genes were characterized to validate our method using Sanger sequencing. Sensitivity, specificity and accuracy reached 93.66%, 99.98% and 99.97 % respectively.
- 2) From 2017 to 2019, Geneplus-Beijing Institute has participated in 14 laboratory quality evaluations and certified by Chinese and international agencies including College of American Pathologists (CAP), the European Molecular Genetics Quality Network (EMQN), National Center for Clinical Laboratory External quality control etc.
- 3) In 2018, as one main unit, Geneplus-Beijing Institute participated in the establishment of a standard database based on the interpretation of genetic variation in Chinese population, led by the China food and drug institute^[10] with the goal to establish a reference system for performance evaluation of BRCA genetic testing and variant interpretation to facilitate clinical decision making. All 53 BRCA1/2 germline mutations identified in cell lines and clinical cases detected by our NGS assay were successfully validated by Sanger sequencing.

6. How did the authors calculate tumor mutational burden?

Author's response: We thank reviewer for this question. The mutation burden was calculated as the number of non-silent somatic mutations (non-synonymous SNV, indel and splice \pm 2) per mega-base (1MB) of genomic regions sequenced (1MB for our panel). CNV or structural variation was not included for mutation burden calculation. To avoid false negative results due to the sample quality that may lead to biased TMB calculation, only tumor DNA specimens with at least one mutation with VAF>3% and ctDNA specimens with at least one mutation with VAF>0.5% were included for the TMB calculation. Other specimens were classified as TMB-unevaluable. We agree that this information will be informative for readers and therefore added these details to the method section in the revised manuscript (Page 13, line 17-line 24).

7. Fig. 3a – “Alternations” should be “alterations”

Author's response: We appreciate the reviewer pointing out this typo. We have corrected it and we have also done thorough proofreading for typos and errors.

8. The authors could also provide some commentary on TP53 mutations and earlier age of diagnosis? Was the difference in age of onset significant for TP53 mutations? Are there specific TP53 variants that predict earlier onset more than others?

Author's response: We appreciate this constructive comment. It is well documented that in Li-Fraumeni syndrome, germline mutations in TP53 were associated with early onset of various cancers^[11, 12]. It is therefore interesting to see whether germline TP53 mutations are associated with sporadic lung cancer in Asian patients. In our cohort, 5 patients were identified with germline TP53 mutations. The median age was 43 years old (range 40-68), younger than 61 years old in patients with other germline mutations although the difference did not reach statistical difference ($p=0.16$) likely due to small sample size. Similarly, there was a trend that patients with germline TP53 germline mutations were younger than patients without P/LP germline mutations (median age 43 vs. 60 years old, $p=0.07$). These data suggest that as the reviewer suspected, TP53 germline mutations may be associated with younger age of cancer onset. We agree with the reviewer that this information would be interesting to the readers. Therefore, we added these results in the revised manuscript (page 6, line 39 - page7 line 2, and page 9 line 13- page 18).

Regarding specific TP53 codons, it is another very good suggestion as it was reported particular TP53 mutations such as p.R158H were identified in different families with Li-Fraumeni Syndrome suggesting certain germline TP53 mutations may be associated with higher risk of cancer develop^[13]. However, the 5 mutations were identified in 5 different codons in our study. Therefore, we were not able to address the question whether certain specific TP53 germline mutations are associated with younger age of onset. We thank the reviewer again for the insightful comment.

9. One obvious caveat of this study is the ACMG classification that was used to determine the pathogenic potential of germline mutations. Authors also have identified this. Can the authors offer a potential strategy to address this in the future?

Author's response: We thank the reviewer for this constructive advice. As the reviewer pointed out, one caveat was that the P/LP germline mutations were defined based on ACMG, which is primarily based on data and experience from Caucasian patients. These rules may not always apply to Asian population. Besides, some potential germline P/LP mutations specific to Asian population were not investigated in the current study. With the intent to fill this void, many efforts have been made. For example, in 2018, as one main unit, Geneplus-Beijing Institute participated in the establishment of a standard database based on the interpretation of genetic variation in Chinese population, led by the China food and drug institute^[10] with the goal to establish a reference system for performance evaluation of BRCA genetic testing and variant interpretation to facilitate clinical decision making for Chinese patients. Future studies on large cohort of Asian patients using more comprehensive panel, even at exome level are warranted to establish clinical germline database for Asian cancer patients. We thank the reviewer again for this constructive comment and we have updated the relevant discussion accordingly (page 10, line 1- line 31).

10. In supplementary table 1 the title mentions 107 patients detected with P/LP mutations, however in the text mentions 106? Correction needed?

Author's response: We thank the reviewer to point out this typo. It should be 106 and it was now corrected in the revised manuscript.

11. There were a number of minor typos throughout the text that should be corrected before publication.

Author's response: We thank you for the meticulous review. We have corrected it and we have also done thorough proofreading for typos and errors.

Reviewer #2 (Remarks to the Author):

This is the first large-scale analysis of germline mutation in lung cancer of Asian. The results are interesting, however, the reviewer feels that more data are needed to understand roles of observed germline mutations.

Author's response: We appreciate reviewer's favorable comments and constructive suggestions.

1. It is unclear whether germline mutations in BRCA1/2 and other homologous recombination-related genes contributed to tumorigenesis or not. Comparison of HRD (homologous recombination deficiency) scores and/or Cosmic signature 3 fractions between P/LP(+) and P/LP(-) populations are needed to draw a conclusion.

2. It is unclear whether the germline mutations in mismatch repair genes contributed to tumorigenesis or not. Comparison of tumor mutation burden as well as Cosmic signature 6 fractions between P/LP(+) and P/LP(-) populations are needed to draw a conclusion.

Author's response: We completely agree with the reviewer that signature analysis could potentially help to solidify the contribution of certain genes, such as BRCA1/2 and mismatch repair (MMR) genes to tumorigenesis. As suggested, we attempted mutational signature analysis. However, the numbers of mutations in each specimen from the cancer gene panel sequencing were too small for reliable signature analysis. In order to address this question with limited data from this study, we combined SNVs from all tumors with germline BRCA1/2 mutations (BRCA group) versus tumors with germline MMR (MLH3, MSH6, PMS1 and PMS2) mutations (MMR group) respectively. With this caveat, we observed COSMIC mutational signature AC3 (associated with BRCA mutations) contributed to 33.5% of SNV in BRCA group vs 20.1% in MMR group. On the contrary, the contribution of two signatures associated with MMR defect AC20 and AC26 accounted for 28.9% in SNVs in MMR group compared to 10.5% in BRCA group (**Reviewer Figure 4**).

Reviewer Figure 4. COSMIC mutational signatures in samples from patients with BRCA1/2 germline mutations versus MMR gene mutations. The stacked bar plot represents fraction of mutations associated with each signature. The mutational signature sets are based on combined Alexandrov and COSMIC signatures (AC1-30)^[14] used in “YAPSA” package.

Homologous recombination (HR) deficiency (HRD) score suggested by the reviewer is an excellent tool to assess the contribution of HR genes to mutagenesis. HRD score is calculated based on SigMA that can potentially derive mutational Signature 3 using limited numbers of mutations from panel sequencing^[15]. As the reviewer suggested, we attempted calculating the HRD scores by SigMA. However, there is currently no model-curated HRD scoring system based on WES/WGS data available for lung cancers so we could not calculate the final HRD scores. Instead, we compared the 3 intermediate parameters that the HRD scores are derived from. As shown in **Reviewer Table 1 (RT1)**, all 3 parameters were higher in specimens with P/LP HR repair (HRR) germline mutations, however the difference was not significant.

Reviewer Table 1. SigMA analysis on lung cancer samples with germline HRR mutations versus other germline mutations

	P/LP HRR (n=49)	P/LP non-HRR (n=32)	P-value
		Signature_3_ml	
Median	0.09872	0.05819	0.27
		Signature_3_c	
Median	0.3328	0.3123	0.47
		Exp_sig3	
Median	0.4399	0	0.35

P/LP HRR: samples with P/LP germline mutations in HR repair genes. P/LP non-HRR; samples with other P/LP germline mutations. Samples with at least 5 somatic mutations found were included.

We also compared the tumor mutation burden (TMB) as suggested and the median TMB was 7 mutations/MB in tumors with P/LP MMR germline mutations, higher than 5 mutations/MB in tumors without germline or somatic mutations in MMR genes, but the difference was not significant (p=0.36). Because some critical clinical information such as smoking status was unknown from a big proportion of this real-world patient cohort, it is difficult to interpret the data.

Once again, we thank the reviewer for this critical point. There was some trend indicating that these germline mutations may have contributed to the somatic mutational processes. However, our analyses had substantial caveat to appropriately address this question. We discussed about this limitation in the revised manuscript (Page 10, line 39 to page 11 line 3).

3. It is unclear whether the germline mutation frequencies in lung cancer patients are higher than those of non-lung cancer population? So, etiologically, the results do not tell a lot for lung carcinogenesis.

Author's response: The authors appreciate this critical and constructive question. To address this question, we derived the P/LP mutations from a recently published whole genome sequencing data of non-cancer individuals enrolled in the China Metabolic Analytics Project (ChinaMAP) (n=10,588)^[16], a study on the impact of genetic architecture on metabolic diseases. Based on the same 2015 ACMG guideline under the same filtering criteria, 63 P/LP germline mutations from ChinaMAP cohort were identified in the 35 cancer predisposition genes identified in the current study, significantly lower than that in the lung cancer cohort (63/10,588 (0.60%) vs 111/1,794

(6.1%), $p < 2.2e-16$, Chi-square test). We then compared the allele frequency (AF) of the germline mutations of each gene in lung cancer patients to that in the non-cancer individuals. There were 17 genes with P/LP germline mutations detected in ≥ 3 patients in the current lung cancer cohort. As shown in the **reviewer table 2**, 16 of the 17 genes had AF of P/LP germline mutations higher (significantly higher in 11 genes) in the lung cancer patients than in non-cancer individuals in ChinaMAP study indicating an enrichment of these germline mutations in lung cancer patients.

Reviewer Table 2. The enrichment of P/LP germline mutation in lung cancer cohort.

Genes	LC AC [#]	LC AN	LC AF	ChinaMAP AC [#]	ChinaMAP AN	AF	OR	95% CI	p-value
BRCA2*	14	3574	0.0039	16	21160	0.0008	5.18	2.34-11.33	2.56E-05
FANCA*	9	3579	0.0025	4	21172	0.0002	13.30	3.71-59.17	1.1440E-05
RAD51D*	7	3581	0.0020	16	21160	0.0008	2.58	0.90-6.65	0.03921
ATM*	7	3581	0.0020	7	21169	0.0003	5.91	1.77-19.76	0.0018
MUTYH	6	3582	0.0017	41	20864	0.0019	0.86	0.30-2.05	1
TP53*	5	3583	0.0014	2	21174	9.445E-05	14.77	2.42-155.07	0.0010
FANCD2*	5	3583	0.0014	2	21174	9.445E-05	14.77	2.41-155.07	0.0010
BRIP1*	5	3583	0.0014	2	21174	9.445E-05	14.77	2.41-155.07	0.0010
BLM	5	3583	0.0014	11	21165	0.0005	2.68	0.73-8.39	0.0699
BRCA1	4	3584	0.0011	12	21164	0.0006	1.97	0.46-6.50	0.2741
MSH6*	4	3584	0.0011	3	21173	0.0001	7.88	1.33-53.73	0.0107
PMS2	3	3585	0.0008	7	21169	0.0003	2.53	0.42-11.09	0.1667
PMS1*	3	3585	0.0008	0	21176	0	Inf	2.44-Inf	0.0030
SDHA	3	3585	0.0008	5	21171	0.0002	3.54	0.55-18.21	0.0968
RAD51C*	3	3585	0.0008	2	21174	9.445E-05	8.86	1.01-106.12	0.0242
PALB2	3	3585	0.0008	6	21170	0.0003	2.95	0.48-13.83	0.1301
CHEK2*	3	3585	0.0008	1	21175	4.722E-05	17.72	1.42-924.85	0.0108

LC: lung cancer, AC: Allele counts (mutant allele count of P/LP mutation found from lung cancer cohort in certain genes), AN: Allele number (Total allele count minus mutant AC in the cohort or database), AF: allele frequency, OR: Odds Ratio, CI: Confidence interval, Inf: infinity. *P-value < 0.05 (calculated with Fisher's exact test). [#]The same ACMG guideline and same criteria were applied to annotate P/LP mutations.

Interestingly, *MUTYH*, a gene encoding a DNA glycosylase involved in oxidative DNA damage repair and associated with heritable predisposition to various cancers, particularly colorectal cancer^[17, 18], demonstrated similar AF between lung cancer patients in this cohort and non-cancer individuals from ChinaMap study suggesting that the germline mutations in *MUTYH* are not specifically associated with lung cancer risk. It is worth noting that an *MUTYH* variant c.934-2A>G (rs77542170) is defined as a "P/LP" mutation based on ACMG guideline. Indeed, the AF of *MUTYH* c.934-2A>G in non-cancer individuals from the Genome Aggregation Database (GAD) was only 0.11% (allele count (AC): 312 of 28,2820) in line with this annotation. However, 307 of the 312 AC were from the East Asians (EAS) with an AF of 1.5% (307/19952) in EAS, 13.6 times higher than that of the whole GAD population ($P = 2.2E-16$). In addition, there were 5 EAS individuals harboring homozygous alleles of

this variant (gnomAD, <https://gnomad.broadinstitute.org>). Furthermore, two studies on Japanese patients reported that AF of this *MUTYH* c.934-2A>G variant in gastric cancer patients^[19] and colorectal cancer patients^[20] was no different compared to non-cancer individuals. These results suggested that *MUTYH* c.934-2A>G (rs77542170) most likely is not a pathologic germline mutation for EAS individuals highlighting the profound impact of ethnicity on the defining P/LP germline mutations.

We sincerely appreciate reviewer for this constructive advice and we think these additional analyses would strengthen the etiologic connection between germline mutations and lung carcinogenesis and emphasize the importance of taking ethnicity into consideration when defining P/LP germline mutations. We therefore included **Reviewer Table 2** (now as **Table 2**) and updated the results and discussion in the revised manuscript. (page 5, line 13 - line 27 in result, page 10, line 35 - line 39 in discussion, page 12, line 17 - line 22 in methods).

4. How the authors declare that the germline mutation-positive patients are genetically ordinal Chinese? Is it possible to perform principle component analysis using SNP information obtained by NGS analysis?

Author’s response: We thank the reviewer for this critical question. As suggested, we compared the SNP data from the current cohort to the SNP of populations submitted to 1000 genomes project phase 3 data (n=2,054)^[21, 22]. First, we calculated the mean pairwise Fst difference between different populations with EIGENSOFT (v7.1.2) and found obvious difference between the patients in the current lung cancer cohort and African (Fst = 0.07), European (Fst = 0.06), South Asian (Fst = 0.04) and Admixed American (Fst = 0.04) populations. However, the SNP architecture of these lung cancer patients was almost identical to that of East Asian population (Fst=0.00) (**Reviewer Figure 5**). Furthermore, the principal component analysis (PCA), as suggested by the reviewer using SNP from 1000 genome project by PLINK^[23] demonstrated that the lung cancer patients from this study were significantly clustered with East Asians (ASN) but not with other populations (**Reviewer Figure 6**). Since all these patients were from China, we’re pretty confident that these patients are most likely genetically ordinal Chinese. We added these data in the revised manuscript (page 4, line 25-37 and page 12 line 24 - line 35).

Reviewer Figure 5. Pairwise Fst difference between the patients in the current lung cancer cohort and other populations in 1000 genome project. AFR: African; AMR: Admixed American; ASN: East Asian; EUR, European; SAS: South Asian.

Reviewer Figure 6. PCA of patients in the current lung cancer cohort and other populations in 1000 genome project. AFR: African; AMR: Admixed American; ASN: East Asian; EUR, European; SAS: South Asian.

References

- [1] Nong J, Gong Y, Guan Y, et al. Circulating tumor DNA analysis depicts subclonal architecture and genomic evolution of small cell lung cancer. *Nat Commun.* 2018. 9(1): 3114.
- [2] Zhang Y, Chang L, Yang Y, et al. The correlations of tumor mutational burden among single-region tissue, multi-region tissues and blood in non-small cell lung cancer. *J Immunother Cancer.* 2019. 7(1): 98.
- [3] Watanabe M, Hashida S, Yamamoto H, et al. Estimation of age-related DNA degradation from formalin-fixed and paraffin-embedded tissue according to the extraction methods. *Exp Ther Med.* 2017. 14(3): 2683-2688.
- [4] Do H, Dobrovic A. Sequence artifacts in DNA from formalin-fixed tissues: causes and strategies for minimization. *Clin Chem.* 2015. 61(1): 64-71.
- [5] Haile S, Corbett RD, Bilobram S, et al. Sources of erroneous sequences and artifact chimeric reads in next generation sequencing of genomic DNA from formalin-fixed paraffin-embedded samples. *Nucleic Acids Res.* 2019. 47(2): e12.
- [6] Hu X, Fujimoto J, Ying L, et al. Multi-region exome sequencing reveals genomic evolution from preneoplasia to lung adenocarcinoma. *Nat Commun.* 2019. 10(1): 2978.
- [7] Lu S, Yu Y, Li Z, et al. Brief Report: EGFR and ERBB2 Germline Mutations in Chinese Lung Cancer Patients and Their Roles in Genetic Susceptibility to Cancer. *J Thorac Oncol.* 2019 .
- [8] Sun S, Liu Y, Eisfeld AK, et al. Identification of Germline Mismatch Repair Gene Mutations in Lung Cancer Patients With Paired Tumor-Normal Next Generation Sequencing: A Retrospective Study. *Front Oncol.* 2019. 9: 550.
- [9] Guan Y, Hu H, Peng Y, et al. Detection of inherited mutations for hereditary cancer using target enrichment and next generation sequencing. *Fam Cancer.* 2015. 14(1): 9-18.
- [10] Qu S, Chen Q, Yi Y, et al. A Reference System for BRCA Mutation Detection Based on Next-Generation Sequencing in the Chinese Population. *J Mol Diagn.* 2019. 21(4): 677-686.
- [11] Park KJ, Choi HJ, Suh SP, Ki CS, Kim JW. Germline TP53 Mutation and Clinical Characteristics of Korean Patients With Li-Fraumeni Syndrome. *Ann Lab Med.* 2016. 36(5): 463-8.
- [12] Amadou A, Achatz M, Hainaut P. Revisiting tumor patterns and penetrance in germline TP53 mutation carriers: temporal phases of Li-Fraumeni syndrome. *Curr Opin Oncol.* 2018. 30(1): 23-29.
- [13] Mezquita L, Jove M, Nadal E, et al. Brief report: High prevalence of somatic oncogenic driver alterations in non-small cell lung cancer patients with Li-Fraumeni Syndrome. *J Thorac Oncol.* 2020 .
- [14] Alexandrov LB, Nik-Zainal S, Wedge DC, Campbell PJ, Stratton MR. Deciphering signatures of mutational processes operative in human cancer. *Cell Rep.* 2013. 3(1): 246-59.
- [15] Gulhan DC, Lee JJ, Melloni G, Cortés-Ciriano I, Park PJ. Detecting the mutational signature of homologous recombination deficiency in clinical samples. *Nat Genet.* 2019. 51(5): 912-919.
- [16] Cao Y, Li L, Xu M, et al. The ChinaMAP analytics of deep whole genome sequences in 10,588 individuals. *Cell Res.* 2020 .
- [17] Win AK, Dowty JG, Cleary SP, et al. Risk of colorectal cancer for carriers of mutations in MUTYH, with and without a family history of cancer. *Gastroenterology.* 2014. 146(5): 1208-11.e1-5.
- [18] Win AK, Reece JC, Dowty JG, et al. Risk of extracolonic cancers for people with biallelic and monoallelic mutations in MUTYH. *Int J Cancer.* 2016. 139(7): 1557-63.
- [19] Tao H, Shinmura K, Hanaoka T, et al. A novel splice-site variant of the base excision repair gene MYH is associated with production of an aberrant mRNA transcript encoding a truncated MYH protein not localized in the nucleus. *Carcinogenesis.* 2004. 25(10): 1859-66.
- [20] Tao H, Shinmura K, Suzuki M, et al. Association between genetic polymorphisms of the base excision repair gene MUTYH and increased colorectal cancer risk in a Japanese population. *Cancer Sci.* 2008. 99(2): 355-60.
- [21] Clarke L, Zheng-Bradley X, Smith R, et al. The 1000 Genomes Project: data management and community access. *Nat Methods.*

2012. 9(5): 459-62.

- [22] 1000 Genomes Project Consortium, Abecasis GR, Auton A, et al. An integrated map of genetic variation from 1,092 human genomes. *Nature*. 2012. 491(7422): 56-65.
- [23] Purcell S, Neale B, Todd-Brown K, et al. PLINK: a tool set for whole-genome association and population-based linkage analyses. *Am J Hum Genet*. 2007. 81(3): 559-75.

REVIEWERS' COMMENTS:

Reviewer #1 (Remarks to the Author):

Dr. Peng and colleagues have addressed all of my initial concerns through the additional revisions. One new note: It was unclear to me (in fact I missed it completely) that there were samples from ctDNA instead of solid tumor FFPE. Please update the text and methods to describe the different liquid biopsy vs. FFPE biopsies. As the genomic methods must have differed substantially between these sample types, please describe the methods for sample processing and library prep and genomic analysis in a way that is specific to each individual sample type.

For ctDNA, was total tumor content quantified and were any filters applied to remove samples without ctDNA?

p.13 line 8 – please clarify the statement that “all mutations were manually verified in IGV browser.” This seems impossible based on the number of patients and mutations. Which mutations were manually verified? Does this statement specifically refer to the P/LP variants?

The supplementary data might be moved to the main figures, since there are only 3 main figures.

Reviewer #2 (Remarks to the Author):

The authors performed additional analyses requested by the reviewer except for mutation signature and HRD analysis. The last two data would be obtained by performing whole exome analysis. However, the present data have almost demonstrated the significance of germline mutations in ordinary Chinese.

1. The present supplementary 2 should be presented as a main Table, since the data indicates significance of detected germline mutations. The results should be also explained in the abstract, since the data is a strong evidence of contribution of germline mutations to lung carcinogenesis and have not been presented by TCGA. Instead, Table 2 may be suitable as a supplementary material. Contents of Figure 3b and Tables 3 and are overlapping, so, should be more summarized.

2. Reviewer Figure 4 should be presented as a supplementary Figure, since the data also support significance of detected germline mutations. The data could be cited in the discussion section.

Point-by-point response to reviewers' comments

Reviewer #1 (Remarks to the Author):

Dr. Peng and colleagues have addressed all of my initial concerns through the additional revisions. One new note: It was unclear to me (in fact I missed it completely) that there were samples from ctDNA instead of solid tumor FFPE.

Please update the text and methods to describe the different liquid biopsy vs. FFPE biopsies. Please describe the methods for sample processing and library prep and genomic analysis in a way that is specific to each individual sample type.

Authors' response: We thank the reviewer for this constructive advice and we described the methods for FFPE samples and cfDNA samples separately in the revised manuscript, which now reads as the following.

Methods:

"Sample processing, DNA extraction and Quantification.

Liquid biopsy samples (peripheral blood, ascitic effusion, pleural effusion, pericardial effusion and cerebrospinal liquid) were collected in Streck vacutainer tubes (Omaha, NE) and processed within 48 hours to separate the supernatant by centrifugation at 1,600 g for 10 min. Buffy coat from peripheral blood was kept for DNA extraction as germline control. The supernatant was transferred to microcentrifuge tubes, and further centrifuged at 16,000 g for 10 min to remove remaining cell debris. Separated liquid biopsy samples and buffy coat were stored at -80°C until DNA extraction.

Separated liquid biopsy samples were isolated for cell free DNA (cfDNA) using a QIAamp Circulating Nucleic Acid Kit (Qiagen, Hilden, Germany). Buffy coat DNA and FFPE tumor tissue DNA were extracted using the DNeasy Blood & Tissue Kit (Qiagen). DNA concentration was measured using Qubit fluorometer 3.0 (Life Technologies) and the Qubit dsDNA HS (High Sensitivity) Assay Kit (Invitrogen,

Carlsbad, CA, USA). The cfDNA size distribution was evaluated using an Agilent 2100 BioAnalyzer and a DNA HS kit (Agilent Technologies, Santa Clara, CA, USA).

The sample quality was assessed based on the following criteria: total amount ≥ 30 ng for cfDNA samples or ≥ 100 ng for tumor FFPE samples; fragment length for cfDNA samples was distributed with a dominant peak at 170 bp proximately^[1, 2].

Library construction, target enrichment and Sequencing.

Before library construction, DNA from buffy coat peripheral blood lymphocytes (PBL) or from FFPE samples was sheared to 200~300-bp fragments using a Covaris S2 ultrasonicator (Covaris, Woburn, MA, USA). Indexed Illumina next-generation sequencing (NGS) libraries were prepared from PBL DNA, tumor DNA, and liquid biopsy DNA using the KAPA Library Preparation Kit (Kapa Biosystems, Wilmington, MA, USA).

The region of frequently mutated 1021 genes (Supplementary data 3) in solid tumors were enriched using a custom SeqCap EZ Library (Integrated DNA Technology, Coralville, IA, USA). Captured hybridization was performed using the manufacturer's protocol. Following hybrid selection, the captured DNA fragments were amplified and then pooled to generate several multiplex libraries.

Of note, for liquid biopsy samples, the duplex sequencing based on a unique identifier tag (UID) were applied to filter repeatedly errors in the consensus bidirectionally and rectify sequencing errors mostly introduced by PCR/sequencing and modify the base quality.

Finally, the libraries were performed on NovaSeq6000 or Hiseq3000 Sequencing System (Illumina, San Diego, CA) with 2×101 -bp paired-end reads. The TruSeq PE Cluster Generation Kit V3 and TruSeq SBS Kit V3 (Illumina, San Diego, CA, USA) were used according to the manufacturer's recommendations." (line 453 –line 491)

1. For ctDNA, was total tumor content quantified and were any filters applied to remove samples without ctDNA?

Authors' response: We thank the reviewer for this important question regarding the quality of ctDNA samples, which is critical for ctDNA associated genomic analysis. As a component of cell free DNA (cfDNA), circulating tumor DNA (ctDNA) originates from tumor cells. Currently, there are no appropriate methods to reliably quantify the proportion of ctDNA in cfDNA specimens. Although the variant allele frequencies of mutations may serve as surrogates for cancer cells theoretically, these are confounded by many factors such as how many genes are covered by the sequencing panel utilized and whether the mutations are clonal present in all cancer cells or subclonal that are only present in a small proportion of cancer cells.

In our study, stringent quality control (QC) and specific filtering criteria were applied at each step of ctDNA analysis, which are now included in the revised manuscript read as the following.

Method:

"The sample quality was assessed based on the following criteria: ≥ 30 ng for cfDNA samples and ≥ 100 ng for tumor FFPE samples as determined by a Qubit fluorometer 3.0 (Life Technologies); fragment length for cfDNA sample was distributed with a dominant peak at 170 bp proximately^[1, 2].

...

Of note, for liquid biopsy samples, the duplex sequencing based on a unique identifier tag (UID) were applied to filter repeatedly errors in the consensus bidirectionally and rectify sequencing errors mostly introduced by PCR/sequencing and modify the base quality." (line 468–471, line 484-487)

*"...Somatic mutations that meet the following criteria were included for further analyses: sequencing average depth $>100\times$ in germline DNA, $>500\times$ in tumor DNA (**1000 in ctDNA**); minimal variant allele frequency (VAF) $> 1\%$ in tumor DNA (**0.5% in ctDNA**); the ratio of allele frequency in case/control (tumor /germline) $>$*

3 and at least 4 supportive reads (both tissue and ctDNA).” (line 536 – 541)

Furthermore, to avoid false negatives that could confound the analysis, for the correlation between germline mutations and somatic mutational landscape, all mutation-negative specimens including 187 liquid biopsy samples were excluded as illustrated in Figure 1. To make sure this is clear to future readers, we added the following in the revised manuscript.

Result:

“To avoid false negative results ascribed to low tumor content in the specimens, 224 patients (including 187 with liquid biopsy samples and 37 with FFPE samples) without any somatic mutation detected were excluded for this analysis (Fig. 1).” (line 242 – 245)

2. p.13 line 8 – please clarify the statement that “all mutaitons were manually verified in IGV browser.” This seems impossible based on the number of patients and mutations. Which mutations were manually verified? Does this statement specifically refer to the P/LP variants?

Authors’ response: We understand why the reviewer raised this concern. This involves tremendous amount of work, which would not be feasible if just for one research project. In fact, this real-world study was based on clinically reported data, for which both somatic and germline variants were manually verified in IGV to confirm the accuracy before release. These has been done by our qualified clinical staff over 2 years.

3. The supplementary data might be moved to the main figures, since there are only 3 main figures.

Authors’ response: We thank the reviewer for this constructive advice. The two figures of FST and PCA analyses were changed as main figure 2a and 2b in revised manuscript. Previous supplementary figure 3 was added to main figures as figure 3b in the revised manuscript.

Reviewer #2 (Remarks to the Author):

The authors performed additional analyses requested by the reviewer except for mutation signature and HRD analysis. The last two data would be obtained by performing whole exome analysis. However, the present data have almost demonstrated the significance of germline mutations in ordinary Chinese.

Author's response: We are glad that the Reviewer is overall satisfied with our additional analyses. We agree that WES would be more appropriate to address these questions. Unfortunately, this is not practical in this real-world study. We are pleased that reviewer accepted that our data were adequate to demonstrate the significance of germline mutations in ordinary Chinese.

1. The present supplementary 2 should be presented as a main Table, since the data indicates significance of detected germline mutations. The results should be also explained in the abstract, since the data is a strong evidence of contribution of germline mutations to lung carcinogenesis and have not been presented by TCGA. Instead, Table 2 may be suitable as a supplementary material. Contents of Figure 3b and Tables 3 and are overlapping, so, should be more summarized.

Author's response: We thank the reviewer for these constructive suggestions. As suggested, we moved Supplementary Table 2 (Title: The enrichment of P/LP germline mutation in lung cancer cohort) as the main Table 2 and updated this in the abstract in the revised manuscript now reads as the following.

Abstract

"A total of 111 pathogenic or likely pathogenic germline mutations were identified, significantly higher than non-cancer individuals (111/1,794 versus 84/10,588, $p < 2.2e-16$)." (line 64-66)

Table 2 (Title: Comparison of germline mutation prevalence between Chinese lung cancer and TCGA lung cancer cohorts) in previous version is now changed as supplementary Table 4 in the revised manuscript.

For Figure 3b and Table 3 in previous version, we kept Figure 3b (current figure 4b) and changed previous Table 3 as Supplementary Table 8 (Title: Somatic mutation

prevalence in germline mutation cohort) in the revised manuscript.

2. Reviewer Figure 4 should be presented as a supplementary Figure, since the data also support significance of detected germline mutations. The data could be cited in the discussion section.

Author's response: We thank the reviewer for this constructive suggestion. We now included Reviewer Figure 4 as Supplementary Figure 1 and discussed these findings in the revised manuscript, which now reads as the following.

Results:

*“Germline mutations may have impact on mutagenesis of lung cancers
We next sought to explore whether germline mutations could have impacted the mutagenesis in this cohort of lung cancers. Mutational Signature analysis can potentially indicate the contribution of certain genes, such as BRCA1/2 and mismatch repair (MMR) genes to tumorigenesis^[4]. However, the numbers of mutations in each specimen from cancer gene panel sequencing applied were too small for reliable signature analysis. We therefore combined SNVs from all tumors with germline BRCA1/2 mutations (BRCA group) versus tumors with germline MMR (MLH3, MSH6, PMS1 and PMS2) mutations (MMR group) respectively. With this caveat, we observed COSMIC mutational signature AC3 (associated with BRCA mutations) contributed to 33.5% of SNV in BRCA group vs 20.1% in MMR group. On the contrary, the contribution of two signatures associated with MMR defect AC20 and AC26 accounted for 28.9% in SNVs in MMR group compared to 10.5% in BRCA group (Supplementary Fig. 1). These data imply that germline mutations may contribute to the tumorigenesis by inducing related mutation type. However comprehensive studies with data at whole exome sequencing level are warranted to validate these findings.” (line 288-304)*

Methods:

“R package “YAPSA” was applied to deconstruct signatures from combined SNV from each group. Signatures (AC) contributed by over 3% of SNVs were displayed.” (line 553-555)

Figure Legend:

“Supplementary Figure 1. COSMIC mutational signatures in samples from

patients with BRCA1/2 germline mutations versus MMR gene mutations. The stacked bar plot represents fraction of mutations associated with each mutation signature.” (21 line 804-807)

References

- [1] Newman AM, Bratman SV, To J, et al. An ultrasensitive method for quantitating circulating tumor DNA with broad patient coverage. *Nat Med.* 2014. 20(5): 548-54.
- [2] Snyder MW, Kircher M, Hill AJ, Daza RM, Shendure J. Cell-free DNA Comprises an In Vivo Nucleosome Footprint that Informs Its Tissues-Of-Origin. *Cell.* 2016. 164(1-2): 57-68.
- [3] Bustin SA, Benes V, Garson JA, et al. The MIQE guidelines: minimum information for publication of quantitative real-time PCR experiments. *Clin Chem.* 2009. 55(4): 611-22.
- [4] Alexandrov LB, Nik-Zainal S, Wedge DC, Campbell PJ, Stratton MR. Deciphering signatures of mutational processes operative in human cancer. *Cell Rep.* 2013. 3(1): 246-59.